


# Observations of wind farm wake recovery at an operating wind farm

Raghavendra Krishnamurthy[1], Rob K. Newsom[1], Colleen M. Kaul[1], Stefano Letizia[2], Mikhail Pekour[1], Nicholas Hamilton[2], Duli Chand[1], Donna Flynn[1], Nicola Bodini[2], Patrick Moriarty[2]

[1]Pacific Northwest National Laboratory, Richland, 99352, United States of America
[2]National Renewable Energy Laboratory, Golden, 80401, United States of America

*Correspondence to*: Raghavendra Krishnamurthy (raghu@pnnl.gov)

**Abstract.** The interplay of momentum within wind farms significantly influences wake recovery, affecting the speed at which wakes return to their free-stream velocities. Under stable atmospheric conditions, wind farm wakes can extend over considerable distances, leading to sustained vertical momentum flux downstream, with variations observed throughout the
diurnal cycle. Particularly in regions such as the US Great Plains, stable conditions can induce low-level jets, impacting wind farm performance and power output. This study examines the implications of wake recovery using long-term observations of vertical momentum flux profiles across diverse atmospheric conditions. In these observations, several key findings were observed, such as a) low-level jet heights are altered downstream of a wind farm, b) a notable impact of low-level jet height on wake recovery is observed using momentum flux profiles at upwind and downwind location, c) detection of wake effects
is almost always observed throughout the atmospheric boundary layer height, and finally d) enhancement of wake recovery is observed in the presence of propagating gravity waves. These insights deepen our understanding of the intricate dynamics governing wake recovery in wind farms, advancing efforts to model and predict their behaviour across varying atmospheric contexts. In addition, the performance of large-eddy simulation-based semi-empirical internal boundary layer height model estimates incorporating real-world atmospheric and turbine inputs was evaluated using observations during low-level jet
conditions.

## 1 Introduction

Wind turbine wakes, i.e., velocity deficits due to extraction of kinetic energy from an operating wind turbine, are observed to extend several kilometres during stable atmospheric conditions both onshore and offshore (Hirth et al., 2012; Banta et al., 2015; Krishnamurthy et al., 2017; Fernando et al., 2019; Ahsbahs et al., 2020; Zhan et al., 2020). Wind farm
wakes from a large cluster of wind turbines in mesoscale model simulations can reach over 50 km downwind under stable atmospheric conditions (Lundquist et al., 2019). Large-eddy simulations of large wind farms also show that wind farm wakes can alter the surface momentum and heat fluxes (Calaf et al., 2011). Wind farm wakes are known to impact local meteorological conditions by, for instance, increasing or decreasing the temperature and enhancing turbulence downwind of a wind farm (Baidya Roy et al., 2004; Smith et al., 2013; Siedersleben et al., 2018; Miller and Keith, 2018, Bodini et al.,





2021), although the intensity of impact depends on atmospheric stability, local atmospheric processes, orientation of the wind farms, downwind distance, number and operative regimes of wind turbines, etc.

In operational wind farms, intra-farm wakes can result in significant power losses and it is important to understand the dissipation of wakes within a wind farm.  The effect of a wind turbine is to decrease the mean velocity and increase the
turbulent kinetic energy above the rotor layer (VerHulst and Meneveau, 2014). The turbulent transport term in a steady-state filtered-energy equation includes the entrainment of mean momentum due to turbulence $[\bar{u}_i\overline{u_i'u_j'}]$ and the entrainment of turbulent kinetic energy due to fluctuating velocities $[0.5\overline{u_i'u_i'u_j'}]$ (Allaerts and Meyers, 2017). Downwind of a wind farm the recovery of a wind turbine wake within a rotor layer largely occurs due to enhanced entrainment of vertical momentum flux from the boundary layer (Abkar and Porte-Agel, 2013; Yang et al., 2014; VerHulst and Meneveau, 2014; Abkar and Porte-
Agel, 2014; Lu and Porte-Agel, 2015). The maximum energy produced by a large (>100MW) land-based wind farm is then constrained by the momentum flux between the surrounding atmosphere and the flow within the wind farm. Therefore, measuring the entrainment of mean momentum due to turbulence upwind and downwind of an operational wind farm can provide insight into the momentum balance of wakes within a wind farm.  The momentum balance can be a function of various locally observed atmospheric phenomena, such as low-level jets, gravity waves, high shear/veer events etc.
Atmospheric stability is known to impact the extent of wake propagation downstream (Hansen et al., 2012, Barthelmie et al., 2012, Hirth et al., 2012, Smith et al. 2013, Krishnamurthy et al., 2017, Lundquist et al., 2019).  In conjunction with some of the local atmospheric features, the transfer of momentum within and outside the surrounding wind farm can show drastic spatial and temporal heterogeneity.

Wind turbines operate within the lowest 300 m of the atmospheric boundary layer (ABL).  Although wind farms
operate within 300 m above ground level, their impacts can be observed through the entire depth of the boundary layer. Therefore, to accurately assess such impacts, observations of mean and turbulent characteristics of wind and temperature should extended up to the top of the ABL.  Remote sensing instruments, such as Doppler lidars, are capable of estimating the mean winds over the ABL (Frehlich, 1994, Frehlich, 2001, Peña et al., 2009, Krishnamurthy et al., 2013, Newsom and Krishnamurthy, 2021) as well as turbulence with accuracy comparable to sonic anemometers, which are considered a
standard for atmospheric turbulence measurements (Frehlich and Cornman, 2001, Frehlich et al., 2006, Banakh and Smalikho, 1997, Krishnamurthy et al., 2012, Sathe et al., 2015, Bonin et al., 2017, Wildmann et al., 2019). Certain observational studies have validated the propagation of wakes for long distances downwind (more than 20 rotor diameters [RDs]), using targeted long-range scanning radar measurements (Hirth et al., 2012; Ahsbahs et al., 2020).  But other observations have also shown that wake deficits are small at larger RDs (~26 RDs) downstream of a multimegawatt wind
farm (Smith et al., 2013). As wakes grow downwind of a wind farm, measuring velocity deficits at larger downwind distances can get very challenging, due to small deficits. Therefore, assessing the impact of wakes purely based on wind and turbulence intensity estimates at targeted observational locations, would not provide a good representation of wake dissipation. Large-eddy simulations have previously shown momentum deficit estimates within and above the wind farm





rotor layer at large downwind distances (Stevens et al., 2016, Gadde and Stevens, 2021) provide more realistic information
about the influence of wind farm wakes on the atmospheric boundary layer. Therefore, accurately measuring momentum deficits at various downwind distances of a wind farm, rather than just mean winds and turbulence intensity profiles, might provide a better assessment of wind farm wakes. Recent studies have focused on observing momentum flux variability around a wind farm using in-situ observations on an aircraft (Syed et al., 2023), but there has not been a study, as per the authors knowledge, looking at any systematic and statistically significant trends in vertical momentum flux profiles under a
variety of atmospheric conditions within an operational wind farm. Therefore, it would be essential to know under what atmospheric conditions wakes recover faster, thereby reducing the impact on downwind wind farms or turbines for optimal siting of wind farms/turbines and power production estimates.

In this paper we investigate the momentum balance (primarily mean streamwise momentum flux $[\langle u'w' \rangle]$)
surrounding an operational wind farm near the Atmospheric Radiation Measurement (ARM) program Southern Great Plains (SGP) sites in Oklahoma during various site-specific atmospheric phenomena. Momentum flux profiles from scanning Doppler lidars and surface sonic anemometers are estimated for both upwind and downwind locations relative to the wind farm. These novel observations reveal the temporal and spatial variability of momentum balance within and above the wind farm wakes during regional specific atmospheric conditions. Profiles also show the growth of the internal boundary layer
and allow quantification of the accuracy of current large-eddy simulation-based approximations in estimating the growth of the internal boundary layer (IBL). Theoretical preliminaries are provided in Section 2, details about turbulence retrievals from sonic anemometers and Doppler lidars is provided in Section 3, and information about the field campaign and site characteristics are given in Section 4. Momentum flux profiles upwind and downwind of an operational wind farm during site-specific atmospheric conditions are discussed in Section 5. Wind farm IBL measurements, comparison of data with
models are given in Section 6 and results are summarized in Section 7.

## 2 Mathematical Preliminaries

Wind farms create a step change in surface roughness and when the boundary layer height ($\delta$) is larger than the surface momentum roughness ($z_{0m}$), an internal boundary layer ($\delta_{IBL}$) is developed in the region downstream of the surface
discontinuity (Elliot 1958, Calaf et al., 2013, Stevens et al., 2016, Krishnamurthy et al., 2022). The boundary layer flow is observed to adjust to this new surface condition and grows with downstream distance ($x$). The growth of the internal boundary layer is a function of mean wind, thermal stratification or atmospheric stability, inversion height of the ABL, and surface turbulence characteristics. In the presence of a wind farm, the growth of an internal boundary layer is also a function of the mean wind turbine spacing and characteristics of the wind turbine performance (Calaf et al., 2013, Stevens 2016,
Stevens and Meneveau, 2017). The turbulent entrainment of mean kinetic energy into a wind farm replenishes the wake of a



wind farm and the height of the internal boundary layer can reach up to $\delta$. During stable atmospheric conditions, $\delta_{IBL}$ will grow to reach $\delta$ within a short distance from the leading edge of a wind farm, resulting in a fully developed IBL. While most of the existing studies have been based on high-resolution models, limited long-term observations of $\delta_{IBL}$ growth are available in the literature. Syed et al., 2023 showed spatial variability of momentum flux measurements from in-situ sensors onboard an aircraft upstream, above, and downstream of a wind farm, but did not provide a vertical profile up to the boundary layer. Therefore, estimating profiles of momentum flux up to the boundary layer depth can provide insights into the impact of internal boundary layers on wind farm dynamics.

Dimensional arguments show that at far field, large $x$, as equilibrium conditions prevail, i.e., $u_*^d/u_*^u = \mathcal{F}_1(z_0^u/z_0^d)$, where $u_*^d$ and $z_0^d$ are downwind friction velocity and roughness length, while the superscripts $u$, refers to upwind estimates (Krishnamurthy et al., 2022). In the presence of a wind farm, the model presented in Calaf et al., 2010 assumes two constant stress levels, above ($u_{*,hi}$) and below ($u_{*,lo}$) the wind turbine, with the difference between those momentum layers given as

$$u_{*,hi}^2 = u_{*,lo}^2 + \tfrac{1}{2}c_{ft}(\langle\bar{u}\rangle z_h)^2, \tag{1}$$

where $\langle\bar{u}\rangle z_h$ is the horizontally and time averaged velocity at hub-height, $c_{ft} = \pi C_T \big/ (4S_x S_y)$, $C_T$ is the turbine coefficient of thrust, and lateral and transverse spacing between the wind turbines is given by $S_x$ and $S_y$. Using a logarithmic wind profile formulation ($U(z) = u_* \ln(z/z_0)/k$), where $k$ is von Karman constant (0.4), a relationship for wind turbine roughness height of the wind farm ($z_{0,hi}$) can be estimated (Stevens 2016). Thereby, the growth of the internal boundary layer due to a wind farm can be estimated using (Willingham et al., 2014)

$$\frac{\delta_{IBL}(x)}{z_{0,hi}} = \frac{\delta_{IBL}(0)}{z_{0,hi}} + C_1\left(\frac{x}{z_{0,hi}}\right)^{4/5}, \tag{2}$$

where, $x$ is the downwind distance, $\delta_{IBL}(0)$ is the internal boundary layer height of the wind turbine rotor top at the first row of the wind farm, $z_{0,hi}$ is the surface roughness due to the presence of a wind farm, and $C_1$ is a growth constant (0.28) estimated from large-eddy simulation models (Calaf et al., 2010, Stevens, 2016). The wind farm surface roughness is a function of the upwind surface friction velocity, wind turbine and farm parameters, and inflow mean wind conditions within the wind turbine rotor layer. Existing large-eddy simulations estimates of internal boundary layers have typically used idealized conditions while real-world atmospheric conditions estimate of IBL might differ considerably due to competing atmospheric conditions. Therefore, observations of internal boundary layer growth are important to understand the impact of wind farms on the ABL and thereby the momentum balance within a wind farm and associated wake replenishment.





## 3 Flux estimation algorithms and approach

Herein we provide details of the algorithms used to estimate momentum fluxes from surface-based anemometers and scanning Doppler lidars, and a methodology to estimate the height of the internal boundary layer from upwind and downwind momentum flux profiles.

### 130    3.1 Momentum flux estimates from sonic anemometers

Sonic anemometers are considered a standard for estimating atmospheric turbulence parameters (Wilczak et al., 2001, Wilczak et al., 2019, Fernando et al., 2019). Three-dimensional (3-D) acoustic anemometers provide measurements of winds and temperature at high temporal frequency (>= 20 Hz), which supports calculation of higher order statistics with good accuracy (Cook and Sullivan, 2020). The sign conventions of the 3-D winds vary for different manufacturers and for Gill Sonic anemometers, which were deployed for this project, the sign conventions are defined as positive for upward vertical wind component ($w$) and upward atmospheric fluxes, $u$ wind component (North-South) is positive towards North and $v$ wind component (East-West) is positive towards the West. The raw and flux data files are generated as per Cook and Sullivan, 2020, and contain 30-minutes of post-processed data and estimates of turbulent fluxes. The sonic data is post-processed by first applying a de-spiking procedure (Goring and Nikora, 2002) to remove any data anomalies and a 2-axis coordinate rotation is performed (Wilczak et al., 2001), which ensures $\langle w \rangle = \langle v \rangle = 0$ and $\langle u_s \rangle = U$, where U is the mean wind speed, $u_s$ is the streamwise component and $\langle \cdot \rangle$ is a 30-minute temporal average. To estimate the fluxes, the average of each variable is estimated over a 30-minute (non-overlapping) window and no detrending of the data is performed to estimate the velocity fluctuations. The stress tensor is then computed using 30-minute measurements of velocity fluctuations and assumed to be statistically stationary over the averaging window.

145

### 3.2 Momentum flux profiles from Doppler lidars

A brief description of the method to estimate momentum flux profiles from lidars is given below (Eberhard et al., 1989; Mann et al., 2010). The radial velocity ($v_r$) equation of a Doppler lidar is given by:

$$v_r(R, \theta) = u \sin \varphi \cos \theta + v \sin \varphi \sin \theta + w \cos \varphi, \tag{3}$$

where, $R$ is the range, $\varphi$ is the half-opening angle of the conical scan (30º), $\theta$ is the azimuthal direction of the lidar beam (0 degrees is North), and $[u, v, w]$ are the wind components at each range-gate center. The variance of $v_r$ is given by

$$\sigma^2[v_r(R, \theta)] = \sigma_u^2 \sin^2 \varphi \cos^2 \theta + \sigma_v^2 \sin^2 \varphi \sin^2 \theta + \sigma_w^2 \cos^2 \varphi + 2\langle u'v' \rangle \sin^2 \varphi \cos \theta \sin \theta +$$
$$2\langle u'w' \rangle \cos \varphi \sin \varphi \cos \theta + 2\langle v'w' \rangle \cos \varphi \sin \varphi \sin \theta, \tag{4}$$



where $\langle \cdot \rangle$ represents 30-minute averaging and $[u', v', w']$ are the velocity fluctuations. One can estimate the streamwise momentum flux components ($\langle u'w' \rangle$) by calculating the radial velocity variance in the upwind and downwind directions over 30-minutes (Eberhard et al., 1989; Mann et al., 2010), which is then given by

$$\langle u'w' \rangle = \frac{\sigma^2[v_{r\,down}] - \sigma^2[v_{r\,up}]}{2 \sin 2\varphi}, \tag{5}$$

where, $\sigma^2[v_{r\,down}]$ and $\sigma^2[v_{r\,up}]$ are the radial velocity variances from the nearest downwind and upwind azimuth angles relative to the mean wind direction, respectively and $\phi$ is the mean wind direction. The nearest up and down radial velocities from the azimuth angles are picked for each 30-minute sample and given range-gate wind direction estimate. It can be noted from Eq. 5 that in a positively sheared turbulent flow, $\sigma^2[v_{r\,up}] > \sigma^2[v_{r\,down}]$, i.e., the upwind variances, are typically larger than downwind variances. The effect of measurement volume is not considered in this analysis and has been shown to have a minimal impact on the streamwise momentum flux measurements for Doppler lidars (Mann et al., 2010).

For evaluating the accuracy of the algorithm, continuous Velocity Azimuth Display (VAD) scans at the same elevation and azimuth angles are required to calculate the variance of the radial velocity along each beam. From October 8, 2020, to January 14, 2021, continuous eight-point Planned Position Indicator (PPI) scans ($\Delta az = 45°$ and el = 60°) were conducted at the Atmospheric Radiation Measurement (ARM) Southern Great Plains (SGP) Central Facility to support an ongoing field campaign. The mean wind direction ($\phi$) at each height is calculated using the approach of Newsom et al. (2017), wherein a chi-square distribution is fit to estimate the horizontal wind vector. Each beam was averaged for 6 seconds to provide a robust estimate of radial velocity and study the effect of noise from individual radial velocity measurements (Frehlich et al., 2001). This averaging could underestimate the variance observed by the lidar. During this study, each 360-degree wind profile was completed in ~1 minute. This provided the ability to calculate variance of radial velocity along each beam and the momentum flux profile using Eq. 5. Momentum flux estimates from sonic anemometer data on a 60-m tower were calculated using the eddy-correlation method (Stull, 1988). Along-wind momentum flux ($\langle u'w' \rangle$) estimates from the sonic anemometer at 60 m and lidar at 75 m from southerly wind directions are shown in the Figure 1a.





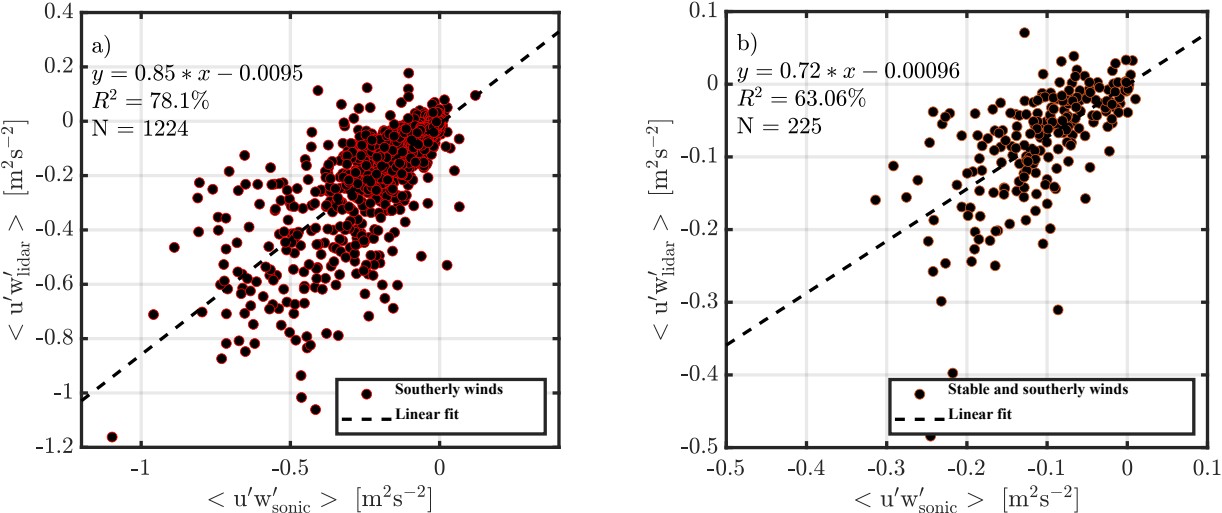

**Figure 1: 30-minute averaged along-wind momentum flux ($\langle u'w' \rangle$) measurements from lidar at 75 m and sonic at 60 m AGL from October 8, 2020, to January 14, 2021 at ARM SGP central facility, for a) southerly wind directions under all atmospheric conditions and b) southerly wind directions under very stable atmospheric conditions (10 *m* < L < 150 *m*). A linear fit between the measurements (y = m\*x + c), coefficient of determination (R$^2$), and number of samples (N) are also shown.**

Doppler lidar $\langle u'w' \rangle$ measurements are observed to correlate reasonably well to sonic anemometer $\langle u'w' \rangle$ measurements, with slope of 0.85 and a coefficient of determination of ~78%. To quantify the effect of stable atmospheric conditions on the accuracy of lidar derived momentum flux estimates, the stability of the atmosphere was determined using the Obukhov length, *L*, given by

$$L = -\frac{u_*^3 T}{kg\langle w'\Theta_v' \rangle}, \tag{6}$$

where *T* is the air temperature, g is the acceleration due to gravity, and $\langle w'\Theta_v' \rangle$ is the kinematic heat flux. During stable atmospheric conditions, given the amount of stratification within the lidar probe volume, the lidar could be measuring very different atmospheric conditions compared to a sonic anemometer. Figure 1b shows measurements from southerly wind directions and very stable atmospheric conditions (10 m < L < 150 m). The coefficient of determination is observed to reduce during stable conditions to ~63%, although the wind speeds are observed to correlate well under all conditions. The transfer of momentum is lowest in stable atmospheric conditions and therefore smaller momentum flux estimates are observed. From a purely statistical standpoint, the smaller magnitude of the fluxes also contributes to reducing the coefficient of determination, since under these conditions the contribution of instrumental and statistical noise to the physical variability of relatively larger. The scatter between lidar and sonic measurements are primarily due to (a) 15 m vertical and ~250 m horizontal separations between sonic anemometer and lidar measurements, (b) low temporal sampling of the lidar measurements, and (c) spatial averaging of the lidar pulse (range-gate = 30 m). These effects amplify during stable atmospheric conditions and result in larger scatter between measurements. Previous observations of momentum flux from profiling Doppler lidars have shown a similar accuracy when compared to sonic anemometers at various heights above





ground level (Mann et al., 2010). Overall, the performance of the algorithm is expected to be adequate for the analysis being conducted in this article.

## 4 Field campaign and site characteristics

Oklahoma ranks third in the United States for installed wind capacity, providing over 37,418 Giga Watt hour (GWh) of electricity in 2022. The state generated approximately 44% of its electricity from wind energy in 2022, the third highest in the country, and provided enough electricity to power the millions of U.S. homes. The landscape and topographic flows around SGP are relatively simple compared to complex terrain sites with low wind speed interannual variability (< 3%) and therefore are favored by wind farm developers (Krishnamurthy et al., 2021). To investigate the interaction between wind farms and the ABL and improve our understanding of wind turbine and wind plant wake effects, the U.S. Department of Energy (DOE) funded a field campaign, American WAKE experiemeNt (AWAKEN), within and adjacent to King Plains wind farm near Enid, Oklahoma (Debnath et al., 2023, Moriarty et al., 2023). Figure 2 shows the domain of the AWAKEN field experiment, various locations with instruments deployed, and operational wind turbines within the domain. Several remote sensing and in-situ sensors were deployed, please see Moriarty et al., 2024 for additional details of the site setup and layout. In this article, data from primarily two instrumented sites are used for data analysis. Site A2 is the inflow site and site H the outflow site to King Plains wind farm during southerly wind directions. Figure 2 shows a picture of both the sites and various instruments deployed. Site A2 was instrumented with a Scanning Doppler lidar, short-range vertical profiling lidar, surface sonic anemometer, and a surface meteorological station, while site H had a scanning Doppler lidar, microwave radiometer, and two disdrometers. In addition, boundary layer height estimates from a ceilometer at site A1 (also an inflow site) were used to evaluate the impact of boundary layer structure on wind farm wake propagation. The wind turbines deployed at the King Plains wind farm are GE 2.8 MW machines with a hub-height of 89 m and a rotor diameter (RD) of 127 m. The average lateral and transverse distance in southerly wind directions between wind turbines (over the Eastern sector of the King Plains wind farm, intersecting sites A1 and H) is approximately 3.15 RD (Sx) and 14.57 RD (Sy). Site A2 is approximately 40 RD upwind of the first row of the King Plains wind farm, site A1 is approximately 2 RD upwind, and site H is approximately 22 RD downwind of the last row of the King Plains wind farm.

Both scanning lidars installed at A2 and H run a composite scan routine that includes 20 minutes of six-beam profiling (Sathe et al., 2015) and 10 minutes of vertical stares. Wind profiles from 100 to 3000 m are obtained by applying the well-established least squares fit to the radial velocity measurement six-beam. Momentum flux is also estimated through the technique described in Sect. 3.2 applied to the upstream and downstream beams based on the selected wind direction sector of interest. In the following, momentum flux measurements from the surface sonic anemometer at the respective site are also combined with the lidar retrieval to extend the observable range down to the surface.





Figure 3a shows the wind rose at 105 m above ground level from Doppler lidar measurements collected from March 17, 2023, to September 10, 2023. Wind directions are predominantly southerly during the duration of the study. Figure 3b shows the distribution of various atmospheric stability conditions as a function of wind direction. Atmospheric conditions were divided into various classes based on the Obukhov length scale as provided in Krishnamurthy et al., 2021. At SGP C1 (which is ~21 km north of King Plains wind farm), stable atmospheric conditions are observed for more than 50% of the time. During neutral conditions, a larger percentage of winds are either easterly or northerly. It is important to note that surface atmospheric stability might not always be representative of conditions at elevated levels, especially during transition periods (i.e., during sunset and sunrise).





**Figure 2. (top)** Location of the AWAKEN field campaign, various sites deployed (yellow stars and circles) during the field campaign, and the wind turbines (black circles) in the area (Moriarty et al., 2024). Images of instruments deployed at site A2 (inflow to King Plains wind farm for dominant southerly wind directions) and site H (downwind of King Plains wind farm) are also shown at the bottom.



**Figure 3. (top)** Wind rose at 105 m AGL from a Doppler lidar at SGP central facility during all atmospheric conditions. **(bottom)** Atmospheric stability classification as a function of wind direction from March 17, 2023, to September 10, 2023. Various atmospheric stability classes are distinguished based on $L$ and defined in Krishnamurthy et al. (2021).

## 4.1 Internal boundary layer estimates from observations



As discussed earlier, an internal boundary layer is developed due to a step change in surface roughness. Turbulence is expected to be higher within the internal boundary layer (downwind of the surface roughness) compared to inflow (upwind of the surface roughness). Wind farms create a step change in surface roughness and are known to develop internal boundary layers downwind of a wind farm (Calaf et al., 2013, Stevens and Meneveau, 2017). In addition to the roughness impacts of the wind turbines, the wind farm developed internal boundary layer is convolved with the wake of the wind

turbines, which create additional momentum deficits downwind of the wind farm. Internal boundary layers can be estimated from a velocity profile, by identifying a kink in the velocity profile (Garratt et al., 1990). Although this method provides a general trend, it is known to be not very accurate. An alternative technique, proposed in Stevens 2016, is being implemented here, where the difference in streamwise momentum flux profiles upwind and downwind of a surface roughness change are used to estimate the growth of the internal boundary layer. Figure 4 shows the median streamwise momentum flux and wind

speed both upwind and downwind of a wind farm from southerly wind directions ($166°< \phi < 190°$) from over 6 months of data collected near an operational wind farm (more details in Section 3). Vertical momentum flux is responsible for the influx of momentum into the wake of a wind farm. Larger streamwise momentum flux deficits above the wind farm are mainly observed due to turbulence and shear generated by the wind turbines. Wind turbine wakes enhance vertical mixing above a wind farm, which results in a downward flow of momentum. The internal boundary layer height ($\delta_{IBL}$) is the height

when the upwind and downwind momentum flux estimates are within 1% of each other above the wind farm. In Figure 4, the $\delta_{IBL}$ is approximately equal to 780 m above ground level (AGL). Therefore, the median impact of the wind farm is observed up to 780 m AGL during southerly wind directions under all atmospheric conditions. Similar technique can be used to estimate internal boundary layer height from models or formulations as shown in Eq 2 above. As previously mentioned $\delta_{IBL}$ can sometimes be inaccurately estimated using the difference between upwind and downwind wind speed

profiles, where $\delta_{IBL}$ is shown to be ~600 m AGL in Figure 4.

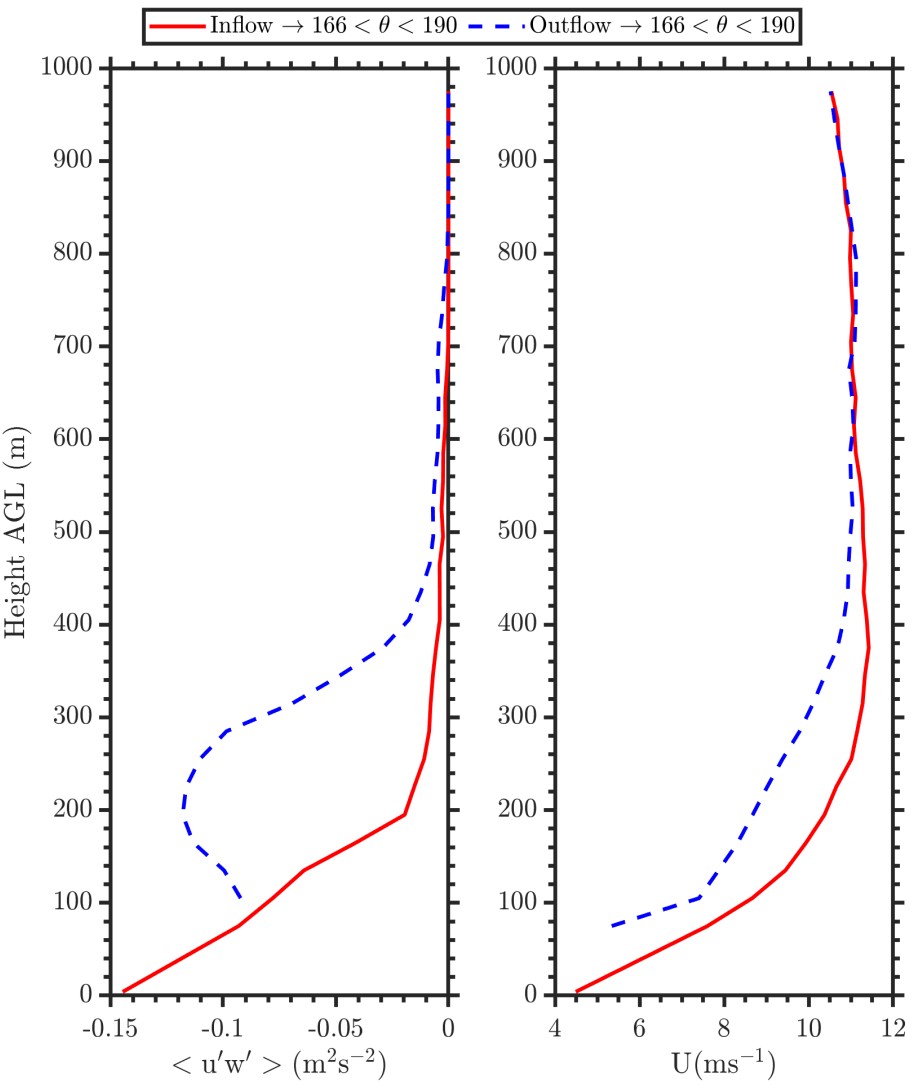

**Figure 4. (left)** Median streamwise momentum flux and **(right)** wind speed profiles site A2 or the upwind site (red) and site H or the downwind site (blue dashed) of the King Plains wind farm in Oklahoma. Only measurements from southerly wind directions ($\phi$) between 166 degrees and 190 degrees are shown.

## 5 Wake Recovery Observations

To minimize the impacts of wakes from neighboring wind farms (including the Breckinridge and Armadillo Flats wind farms shown in Figure 2) in our analysis and to measure the impact of wakes from at least 3 rows of wind turbines, measurements only from southerly wind directions, especially from 166 degrees to 190 degrees are considered. Since the wind directions are predominantly southerly, sufficient data is available to accurately estimate the mean trends of momentum flux during specific atmospheric conditions. Below, statistics of momentum flux variability under different atmospheric



conditions, such as varying levels of thermal stratification (stability), low-level jets, high wind shear and veer conditions,
ABL depth, and atmospheric gravity waves are assessed.

### 5.1 Impact of atmospheric stability on wake recovery

As shown in Figure 4b, when winds are southerly the atmospheric stability is predominantly (>50%) stable near the
surface, with neutral conditions occurring about 20% of the time and unstable conditions observed for the remainder. Figure
5 shows median momentum flux ($\langle u'w' \rangle$) profiles from the downwind (site H) and upwind (site A2) locations during stable,
neutral, and unstable atmospheric conditions. In general, we observe higher negative momentum flux upwind of the wind
farm near the surface with an asymptotic behaviour eventually reaching zero near δ, like a canonical atmospheric boundary
layer. Downwind of the wind farm, enhanced $-\langle u'w' \rangle$ is observed due to the shear and turbulence generated by the wind
turbines. Stable conditions are observed to show larger deviations in momentum flux downwind of the wind farm compared
to neutral or unstable atmospheric conditions. The sign of momentum flux is tied to the vertical wind shear, as for
sustenance of turbulence within a wind farm an increase in wind shear (positive) should result in negative momentum flux
downwind of a wind farm. Therefore, in stable atmospheric conditions, due to large (positive) wind shear, the momentum
flux must be negative to create downwind turbulence. As mentioned earlier, the wind plant wake propagates longer in stable
conditions due to lower ambient turbulence compared to convective conditions, therefore at site A2 which is approximately
22 RD downwind of the last row of the King Plains wind farm, the region of enhanced vertical momentum flux due to the
wind farm is expected to be more persistent and varies with the diurnal cycle (Figure 5a). In Figure 5b and 5c, larger
momentum flux estimates are observed near the surface during unstable and neutral atmospheric conditions compared to
stable conditions. For neutral conditions, since the shear is less positive and higher ambient turbulence compared to stable
conditions is expected, the downwind wind turbine generated momentum flux is expected to be lower or not persistent, and
wakes are not expected to propagate longer distances. But in Figure 5b, significant momentum flux deficits are still observed
at 22 RDs downwind during neutral conditions. A couple of possible reasons for this could be due to a) misclassification of
atmospheric stability from surface flux measurements, i.e., surface atmospheric stability at hub-height is not representative
of the true atmospheric state and b) higher wind shear observed during neutral conditions resulting in more negative
momentum fluxes within the wind farm wake. Larger the vertical momentum flux faster the wake velocity recovers to a
freestream value (Syed et al., 2023). While during unstable atmospheric conditions, wakes are expected to dissipate faster
(convective mixing of the atmospheric boundary layer) and due to low wind shear, the momentum flux deficits are expected
to be significantly lower. In Figure 5c, it is evident that during unstable conditions the momentum flux deficits are lower but
still observed 22 RD downwind. One potential reason for deficits observed during unstable conditions at 22 RDs downwind
could be the impact of conventional updrafts or downdrafts on propagation of wake downwind of a wind farm (Berg et al.,
2017, Wang et al., 2020). Additional analysis is required, ideally using high-resolution large-eddy simulations, to truly
evaluate the impact of updrafts and downdrafts on wind farm wakes. Overall, the median deficit observed over King Plains



wind farm shows that the flow disturbance downwind of a wind farm can extend long distances (at least 22 RDs) in every atmospheric condition. Such differences are generally not very evident from solely observing wind profile observations upwind and downwind of a wind farm.

As mentioned earlier, no observations of vertical profiles of momentum flux have been recorded to date within an operational wind farm, therefore, there is limited knowledge on the height at which the peak transfer of momentum occurs downwind of a wind farm. It is well known that the peak velocity deficit (upwind – downwind velocity) generally occurs at hub-height, but there are no observations showing the peak momentum deficit above the wind farm. Based on large-eddy simulation results, the peak momentum deficit is expected to occur near the upper edge of the wind turbine rotor layer

(Abkar and Porte-Agel, 2015), but based on observations at King Plains wind farm, in stable conditions, at 22 RD downwind, peak momentum flux is consistently observed at ~0.36 RD above the wind farm. Therefore, the mean kinetic energy entrainment height into the wake is observed to occur higher than traditional LES models.  Additional comparisons between LES models and observations are required to further evaluate the wake recovery processes within a wind farm wake.




**Figure 5.** Momentum flux profiles at ~ 40 RD upwind (site A2) and 22 RD downwind (site H) of the King Plains wind farm during
a) stable, b) near-neutral and c) unstable atmospheric conditions. The vertical extent of the wind turbine rotor layer is also shown
with horizontal grey lines. Near surface Obukhov length (L) is used to differentiate between different stability conditions.
Measurements only from southerly wind directions, specifically from 166 deg to 190 deg, and from 17 March 2023 to 10
September 2023 are considered in this analysis.

## 5.2 Impact of low-level jets on wake recovery

Stable conditions produce low-level jets (LLJ) in the U.S. Great Plains (Berg et al., 2015, Krishnamurthy et al., 2021)
whose characteristics can modulate wind farm performance/power output (Gadde and Stevens, 2020). There are several
definitions of low-level jet height ($Z_{LLJ}$) in the literature (Blackadar, 1959, Bonner 1968, Whiteman et al., 1997, Song et al.,
2005, Kalverla et al., 2019, Debnath et al., 2020) but in this article is defined as per Song et al., 2005. The definition is





based off two criteria, 1) wind speed maximum (i.e., low-level jet nose) is observed within the lowest 2-km and is greater

than at least >10 ms$^{-1}$, and 2) wind speed drop-off above the jet-nose is observed and above a set threshold. Three categories

of the LLJs were identified based on varying thresholds of drop-off speeds and maximum nose wind speed (Song et al.,

2005), although in this analysis all LLJ categories were combined. Figure 6a shows the distribution of various near-surface

atmospheric stability classes during southerly wind directions (from 166 deg to 190 deg) and associated median $Z_{LLJ}$ for each

atmospheric stability class. It can be observed that lower $Z_{LLJ}$ are typically associated with very stable or stable near-surface

atmospheric conditions, while higher $Z_{LLJ}$ are observed when the surface atmospheric stability is not stable, indicating a

decoupled boundary layer (Vanderwende et al., 2015). Therefore, there could be confounding influences of the near surface

stability and low-level jet influence on the wind farm wakes during such instances. Figure 6b shows low-level jet nose wind

speed as a function of median $Z_{LLJ}$ per wind speed bin and hub-height wind speed. It is evident that higher the $Z_{LLJ}$, higher

the jet nose wind speed and higher the hub-height wind speed. Overall, it is challenging to decipher various processes

influencing wind farm wake recovery using observations but would be possible to isolate certain common features known to

influence wind farm recovery (such as low-level jet height, or atmospheric stability or atmospheric boundary layer or hub-

height wind speed) and study the variability observed during such select features.

As previously observed using historical measurements from the ARM SGP site, $Z_{LLJ}$ generally falls within 500 m

above ground level (Debnath et al, 2023). Since the scanning lidar measurements start from ~100 m above ground level, in

this analysis we only evaluate low-level jets observed above the rotor layer (> 127 m). Therefore, the observations are

partitioned into two halves, a) 250 m < $Z_{LLJ}$ < 500 m and b) 127 m < $Z_{LLJ}$ < 250 m. Figure 7 shows vertical profiles of

momentum flux and wind speed both upwind and downwind of a wind farm for different low-level jet heights ($Z_{LLJ}$) as

mentioned above and further conditioned to only southerly wind directions (166 deg to 190 deg). During southerly LLJ

events at King Plains wind farm, it is being observed that the transfer of momentum into the wake of the wind farm is a

function of the LLJ height. Higher $Z_{LLJ}$ is associated with larger momentum transfer within the wake and lower velocity

deficit at 22 RD downwind. In short, the wake recovery is faster when the LLJ height is higher. This is mainly due to the

shear generated turbulence below $Z_{LLJ}$ and the enhanced momentum deficit developed due to the wind farm. The peak

entrainment height is observed to marginally increase with higher $Z_{LLJ}$. These results support some of the hypotheses from

previous LES model results on this topic (Gadde and Stevens, 2020). One unique feature that is observed when 127 m <

$Z_{LLJ}$ < 250 m, is that the low-level jet height is modulated due to the presence of a wind farm downwind of the wind farm,

wherein the $Z_{LLJ}$ is consistently observed above the wind farm at 22 RD downwind. The downwind $Z_{LLJ}$ is approximately

equal to the height of the internal boundary layer ($\delta_{IBL}$) generated due to the presence of the wind farm. In addition, during

low-level jet conditions, $Z_{LLJ}$ is typically assumed to be the top of the atmospheric boundary layer ($\delta$, Liu and Liang 2010),

as the turbulence above the $Z_{LLJ}$ is negligible. Figure 8 shows a schematic of the interaction between the wind farm, varying

low-level jet heights and growth of the internal boundary layer due to the presence of the wind farm.


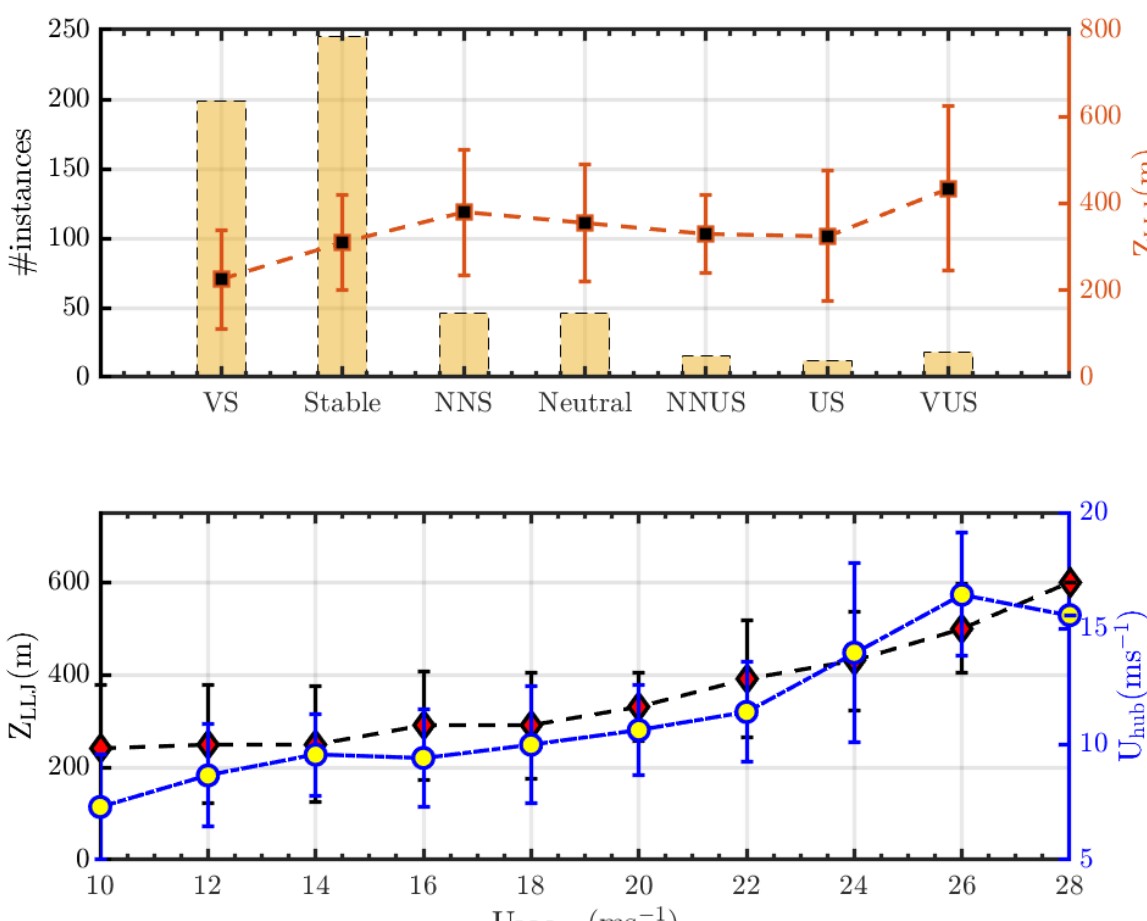

**Figure 6. (top) Distribution of various atmospheric stability classes (VS – Very Stable, Stable, NNS – Near-Neutral Stable, Neutral, NNUS – Near-Neutral Unstable, US – Unstable, VUS – Very Unstable, as per Sathe et al., 2015) and associated $Z_{LLJ}$ per stability class and (bottom) median LLJ nose wind speed ($U_{LLJnose}$) as a function of $Z_{LLJ}$ and Hub-height wind speed ($U_{hub}$) at the upwind site (site A2). The error bars indicate one standard deviation. Minimum $Z_{LLJ}$ is 120 m and maximum $Z_{LLJ}$ is 690 m AGL. Measurements only from southerly wind directions, especially from 166 deg to 190 deg, and from 17 March 2023 to 10 September 2023 are considered in this analysis.**



**Figure 7.** (left) streamwise momentum flux and (right) horizontal wind speed profiles upwind (red, site A2) and downwind (blue, site H) of the King Plains wind farm during conditions when the $Z_{LLJ}$ is less than 250 m AGL at the upwind location (dash-dotted) and $Z_{LLJ}$ is between 250 m and 500 m AGL (star dotted). Measurements only from southerly wind directions, especially from 166 deg to 190 deg, and from 17 March 2023 to 10 September 2023 are considered in this analysis.







**Figure 8. Schematic of impact of low-level jets on wind farm boundary layer when (top) 100 m < $Z_{LLJ}$ < 250 m and (bottom) 250 m < $Z_{LLJ}$ < 500 m.**




### 5.3 Impact during varying shear and veer (non-LLJ) conditions on wake recovery

High wind shear and veer conditions are generally observed within a wind farm, but it is difficult to decouple the effects of shear or veer conditions compared to atmospheric stability conditions. Nonetheless, it would be helpful to observe any consistent trends during such conditions as they are known to impact power production of a wind farm (Murphy et al., 2020).

In addition, such findings may be informative for wind farm control concepts that yaw wind turbines away from the predominant wind direction at hub-height but do not currently consider the wind veer within the rotor layer (Fleming et al., 2019). Figure 9 and Figure 10 show median profiles of momentum flux and horizontal wind speed both upwind and downwind of the King Plains wind farm during non-LLJ events for various shear and veer conditions, respectively. Estimates associated with positive and negative wind shear or veer conditions during southerly wind directions are provided.

Hub-height wind speed ($U(H)$) and shear exponent ($\alpha$) are estimated by fitting a power-law vertical wind profile to the wind speed data available within the rotor layer (90 m to 153 m). The power-law fit is conveniently recast into a linear fit through a log transformation as shown in Eq. 7 below:

$$log\ log\ U\ (z) = log\ log\ U(H) + \alpha\ log\ log\ \left(\frac{z}{H}\right) \tag{7}$$


Where $log\ log\ \left(\frac{z}{H}\right)$ and $log\ log\ U(z)$ are the independent and dependent variables of the linear fit, respectively. Hub-height wind direction ($\phi(H)$) and veer ($\beta \equiv \frac{\partial \phi}{\partial z}$) are estimated in a similar fashion but using a linear wind veer model, as follows:

$$\phi(z) = \phi(H) + \beta(z - H) \tag{8}$$

This approach for estimating hub height quantities has the advantage of leveraging all the available measurements while mitigating possible biases due to the lack of data in the lowest half of the rotor layer.

Certain trends are immediately observed for cases with positive or negative shear conditions (Figure 9), where
regardless of the wind shear profile, momentum deficits generated due to the wind farm are observed 22 RDs downwind of the wind farm. But both wind and momentum deficits are higher during conditions with high wind shear ($0.5 < \alpha < 2$) compared to low or negative wind shear ($\alpha \leq 0$) cases. Although a lower number of cases were recorded when the wind shear was low or negative at the King Plains wind farm compared to high wind shear cases. Median wind speeds at hub-height during both conditions are $\sim 5$ ms$^{-1}$. For cases with negative wind shear, the median wind speed differences between
upwind and downwind are negligible but a clear trend in median momentum flux profiles are observed. This raises an interesting question, according to the authors, what is a good metric to assess the extent of a wind farm wake? Traditionally



it is expected for the wind speed to reach approximately 99% of the free stream wind speed, but in cases such as this where the wind speeds do reach near free stream, it could be erroneously assumed that the wind farm wake has completely been dissipated. In reality, added turbulence due to the presence of a wind farm is not completely removed downwind of a wind farm under such circumstances. Additional research is needed to evaluate wind farm wake models or parameterization schemes during such canonical atmospheric conditions.

Although during cases with positive ($\beta > 0.1^{\circ}\,\mathrm{m}^{-1}$) and negative ($\beta < -0.1^{\circ}\,\mathrm{m}^{-1}$) wind veer conditions, median wind speeds are closer to the cut-in wind speeds of the GE 2.8 MW wind turbines and therefore no momentum flux deficits are observed (Figure 10). Both high and low wind veer cases show that the median wind speeds are relatively low at King Plains wind farm. It is noted that the wind farm has also been observed to modify the wind direction downwind of a wind farm (not shown).



**Figure 9.** Median (left) momentum flux and (right) horizontal wind speed profiles upwind (red) and downwind (blue) of the King Plains wind farm during high (0.5 < α < 2) and negative (α ≤ 0) wind shear conditions. Measurements only from southerly wind directions, especially from 166 deg to 190 deg, and from 17 March 2023 to 10 September 2023 are considered in this analysis.

**Figure 10. (left)** Momentum flux and **(right)** horizontal wind speed profiles upwind (red) and downwind (blue) of the King Plains wind farm during high ($\beta > 0.1^{\circ}\,m^{-1}$) and negative ($\beta < -0.1^{\circ}\,m^{-1}$) wind veer conditions. Measurements only from southerly wind directions, especially from 166 deg to 190 deg, and from 17 March 2023 to 10 September 2023 are considered in this analysis.

### 5.4 Impact of ABL depth on wake recovery

The atmospheric boundary-layer height ($\delta$) is an important parameter for understanding the exchange of momentum, heat, and moisture between the free troposphere and the surface. Unfortunately, estimating $\delta$ can be challenging



and there is limited consensus on the best approach to estimate δ from remote sensing instruments (Kottahaus et al., 2023). Typically, instruments such as scanning Doppler lidars or ceilometers are used to estimate δ (Tucker et al., 2009, Krishnamurthy et al., 2021, Zhang et al., 2022). A ceilometer was deployed at the inflow site A1 and is used to estimate δ

during southerly wind directions for this evaluation. The boundary layer (or mixing) height, provided by the Vaisala CL-31 BL-View software, is based on three different algorithms a) gradient method (where the algorithm detects the gradient in backscatter profile), b) profile method (where the algorithm determines the mixing height by fitting an idealized backscatter profile to the observed range-corrected ceilometer backscatter profiles) and c) merged gradient and profile fit method (Zhang et al., 2022). There are several filters applied to the data, such as cloud and precipitation filters, and additional outlier

removal techniques (due to instrument noise). Figure 11 shows the median momentum flux and horizontal wind speed profiles during southerly wind directions and measurement co-located when concurrent ceilometer measurements were also available. The median δ (~540 m) at A1 from the ceilometer is below the height where both the median momentum flux estimates at sites A2 and H are near equal (~760 m). Therefore, it is observed that the median ceilometer δ does not accurately predict the height of the atmospheric boundary layer. There are several possible reasons for this, for example a)

Ceilometer uses the gradient in backscatter aerosol concentrations to estimate δ, which might not always be the top of the boundary layer (Zhang et al., 2022), and b) upwind and downwind δ might be different due to the presence of the wind farm, therefore the δ estimated by the ceilometer well represents the δ at the inflow site (upwind momentum flux is observed to be close to zero). Above δ, the momentum flux deficits due to the presence of the wind farm are negligible. But overall, it is evident that the impact of the wind farms almost always reaches to the top of the atmospheric boundary layer. Therefore, it

is important to model not only the wind turbine rotor layer with high vertical resolution but up to δ to accurately assess the impacts of wind farms and wake recovery. We further recommend a long-term (1-year +) evaluation of wind farm wake models or parameterizations using long-term atmospheric flux observations through the boundary layer to assess the true performance of such models.

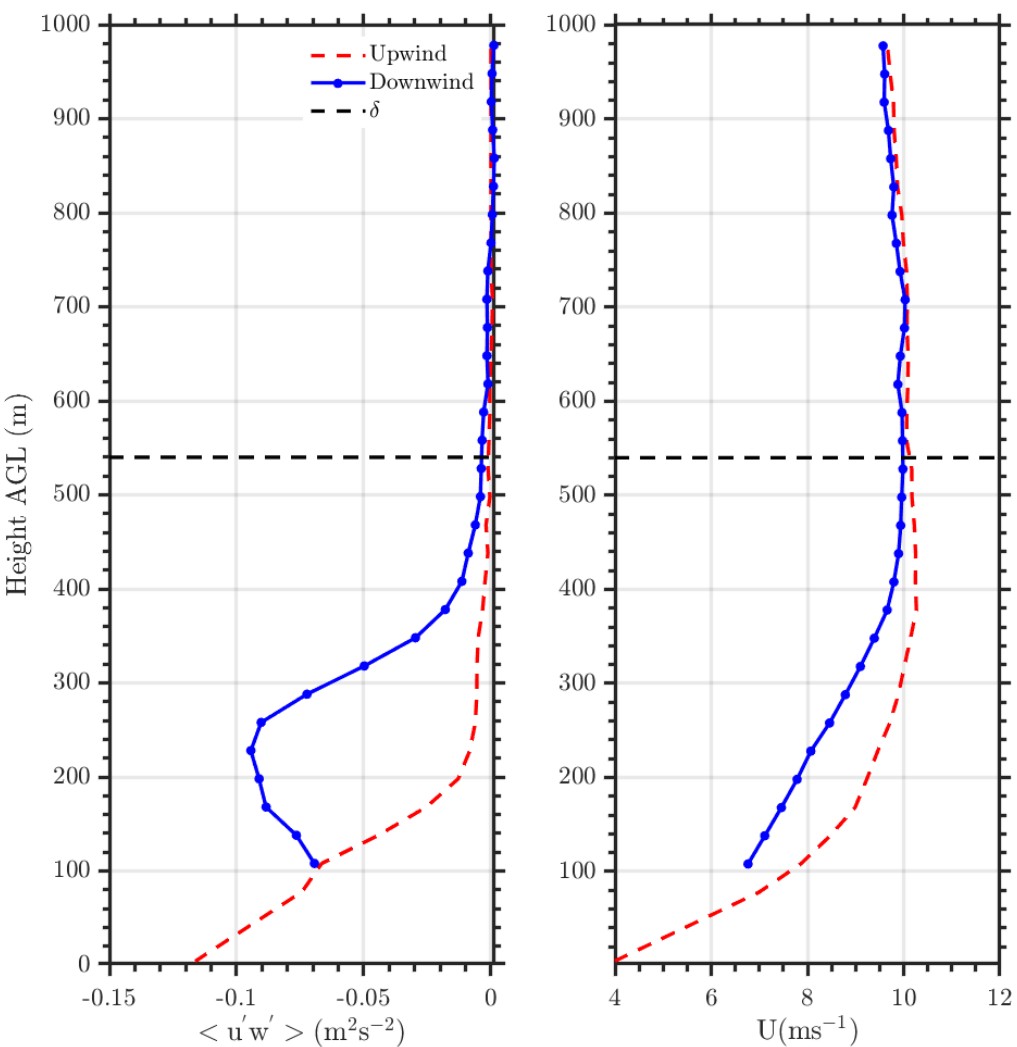


**Figure 11. (left) Momentum flux and (right) horizontal wind speed profiles upwind (red) and downwind (blue) of the King Plains wind farm for all atmospheric conditions. The median height of the ABL (δ) from ceilometer measurements is also shown. Measurements only from southerly wind directions, especially from 166 deg to 190 deg are considered in this analysis.**

**5.5 Impact of gravity waves on wake recovery**

Gravity waves and atmospheric bores are ubiquitous in the SGP region (Carbone et al., 1990, Davis et al., 2003, Geerts et al., 2017). Although most frequently observed during nocturnal and stable atmospheric conditions, they vary significantly in their period and amplitude. These nocturnal convective systems typically accompany high winds, intense rain and/or hail and sometimes tornadoes (Maddox, 1980). The forecast skill of such atmospheric events is relatively low in both numerical

weather prediction models and coarse-grid climate models (Davis et al., 2003, Pritchard et al., 2011). They also typically include a low-level jet within the atmospheric boundary layer, which supports the moisture transport above the stable





boundary layer over SGP (Berg et al., 2015, Krishnamurthy et al., 2022). Primarily gravity waves create wave-like oscillations in the atmosphere due to the presence of a density gradient and bore disturbances are shown to have a significant upward displacement of wind within the troposphere (Rottman and Simpson, 1989, Parson et al., 2019). Such wave-like
disturbances when reaching the surface, can create undulations in the mean winds depending on the period and wavelength of the wave. Mountain waves have previously been known to impact the power production of a wind farm (Draxl et al., 2021) but the impact of propagating gravity waves on wake recovery is not very well understood.

Figure 12 shows a time-height cross-section of vertical velocity as observed near ARM SGP central facility Doppler lidar (Newsom and Krishnamurthy, 2020, Krishnamurthy et al., 2021) on 24th July 2023 from 0300 to 0400 hours UTC
(1900 to 2000 hours local time). The vertical velocity clearly shows wave-like features at approximately 800 m AGL, where we observe a positive and negative shift in vertical velocity. Figure 13 shows median momentum flux and wind speed both upwind and downwind of the King Plains wind farm on 24th July 2023 from 0330 to 0400 UTC. The winds were predominantly southerly during this event with a veer of 0.0375 $^{o}$/m from surface up to the top of the boundary layer. A nocturnal low-level jet was also observed at approximately 220 m AGL at the inflow site and the gravity wave is propagating
above the nocturnal stable atmospheric boundary layer ($\sim Z_{LLJ}$). The peak-to-peak amplitude of the gravity wave is observed to be $\sim$600 m, which spans from 400 m above $\sim$1000 m.

As observed in Figure 13, the momentum flux deficit is significantly enhanced above and within the wake of the King Plains wind farm. Estimates of vertical momentum flux deficits are more than three times compared to median estimates of momentum flux deficits observed during low-level jet conditions (see Figure 7). Downwind, as previously
noted, the peak of a low-level jet is observed to be displaced significantly above the wind farm which is a function of the enhanced mixing within the wind farm rotor layer. The higher the mixing, larger the entrainment, and higher the displacement of the low-level jet. Above the low-level jet, the gravity wave is observed to have an inverse effect, where the momentum flux is positive, entailing momentum is transferred upwards to the gravity wave. The positive or negative transfer of momentum near the gravity wave probably depends on the updrafts or downdrafts of the wave, but over the 30-
min average observations the overall transfer of momentum is upwards. Therefore, further research is needed to understand if gravity waves are being dissipated faster due to the presence of a wind farm. The negative wind shear above the low-level jet could also add to the extraction of momentum from the wind farm. This reduces the intensity of the low-level jet and the extracted momentum results in higher wind speeds above the low-level jet downwind of the wind farm. It is observed that in this case, the wind farm has significantly altered the winds not only within the δ, but also above and modulated the shape and
intensity of the low-level jet. Additional analysis would be needed to see the spatial impact of wind farms during gravity wave propagation and power production. Next steps would be working towards a climatology of gravity waves in the region and its impact on wind farm performance.



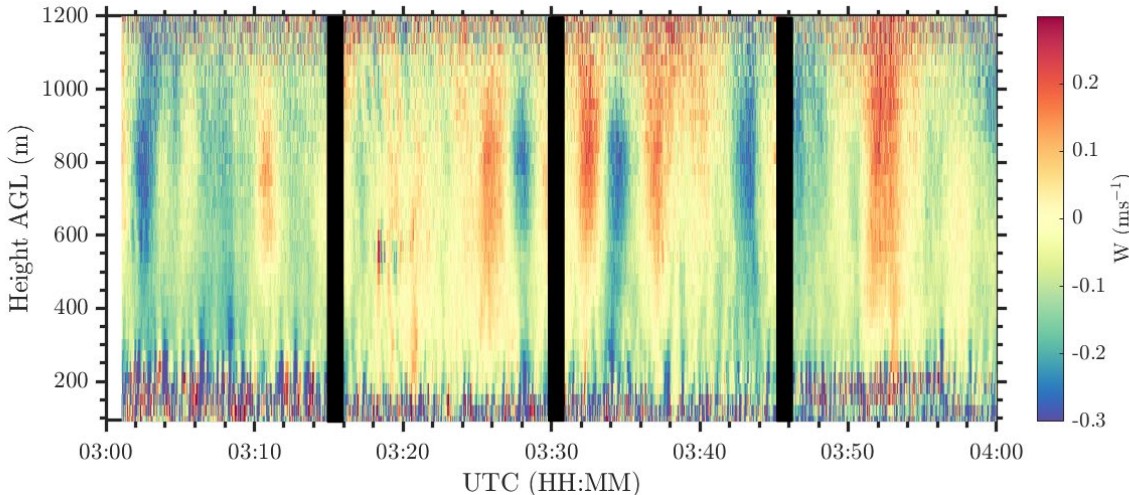


**Figure 12. Time-height cross-section of vertical velocity (W) at ARM SGP central facility on 24-Jul-2023 from 03:00 to 04:00 UTC. The gaps in data every 15-minutes are when the lidar performs wind profiles (Newsom and Krishnamurthy, 2020).**



**Figure 13. (a) Momentum flux and (b) horizontal wind speed profiles upwind (red) and downwind (blue) of the King Plains wind farm during a gravity wave event on 24 July 2023 at 3:30 UTC.**




## 6. Internal Boundary Layer Height

Internal boundary layer height ($\delta_{IBL}$) can be estimated using the difference between inflow and outflow momentum flux
profiles. Wind farm $\delta_{IBL}$ is typically capped by $\delta$ but has been previously shown in large-eddy simulations to penetrate the
upwind $\delta$ during low-level jets (Gadde and Stevens, 2020). As shown earlier, model formulations exist to estimate $\delta_{IBL}$
downwind of a wind farm (Eq. 2). But the growth constant and wind farm surface roughness formulations have been fine-
tuned based on large-eddy simulations, and as per the author's knowledge, no significant analysis on the validation of such
formulations have been conducted so far using real-world observations. This article does not attempt to do that comparison
in detail but is the start of such comparisons. The model formulations (Eq. 2) are sensitive to upwind surface roughness ($z_{0,\,hi}$) estimates and the growth constant ($C$), which can significantly vary $\delta_{IBL}$ estimates within the model (sensitivity analysis
not shown for brevity). The model formulations also do not explicitly restrict the growth of $\delta_{IBL}$ but are implicitly treated in
models where the sub-grid scale mixing does not exceed $\delta$, especially during stable atmospheric conditions.

Figure 14 shows the difference in estimates of $\delta_{IBL}$ from observations and model formulations (Eq. 2) only during stable
atmospheric conditions and in the presence of a low-level jet. Some of the inputs to model formulations are provided by
real-world observations, such as the lidar 10-minute average wind speed at hub-height, inflow surface friction velocity and
roughness from sonic anemometers, thrust curve of the GE 2.8 MW wind turbine, and average wind turbine spacing in
southerly wind directions. Given real-world inputs, the median $\delta_{IBL}$ difference between the model and observations is
approximately 50 m, i.e., the model underestimates the $\delta_{IBL}$. Although there are some extreme cases ($\Delta\delta_{IBL} > 200$ m), where
the model does not behave due to non-consistent wind directions for a sustained period (several hours), the overall
performance of the model can be considered satisfactory. Additional research is needed with large-eddy simulations or
numerical weather prediction models with real-world forcing to assess the recovery and growth of $\delta_{IBL}$ at an operational wind
farm.





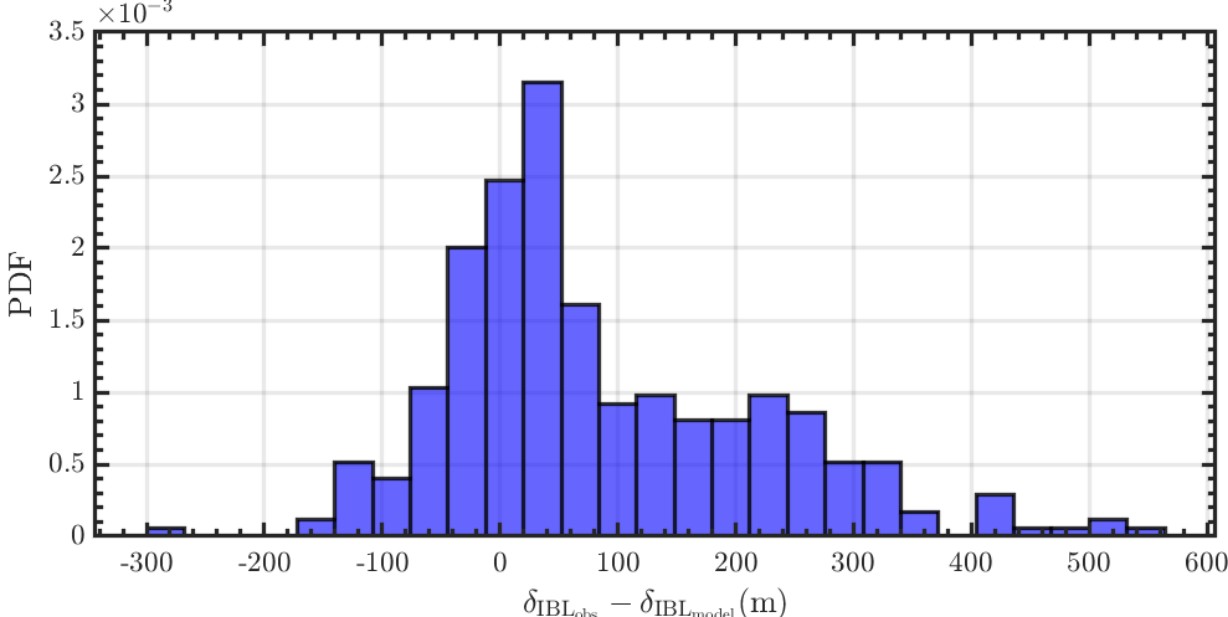

**Figure 14. Probability distribution of $\delta_{IBL}$ difference between observations and a semi-empirical model (Eq. 2) during stable atmospheric conditions and in presence of a low-level jet.**

## 7. Conclusions

The momentum balance surrounding a wind farm can impact wake recovery. The greater the momentum flux the faster the wake velocity recovers to a freestream value. Wind farm wakes are known to propagate long distances during stable atmospheric conditions, and thereby the vertical momentum flux added due to the presence of the wind farm is more persistent at downwind locations and varies with the diurnal cycle. Stable conditions also produce low-level jets in the US Great Plains, which are known to impact the performance and power output of commercial wind farms. Figure 14 shows a schematic of the impact of varying low-level jet height on wind farm wake recovery and modulation of the flow downwind of the wind farm. In addition, presence, and size of an internal boundary layer above the wind farm will impact the momentum exchange between the ABL and the wind farm boundary layer. Therefore, long-term measurements of vertical momentum flux upwind and downwind of a wind farm can provide a holistic view of the physical mechanisms behind wake recovery during various atmospheric conditions. In this study, we have evaluated the impact of wake recovery using long-term observations of vertical momentum flux through the boundary layer during a variety of atmospheric conditions. Overall, some highlights of the observations are mentioned below:

1. Wakes propagate longest during stable atmospheric conditions (see Figure 5),
2. Wind farms alter low-level jet characteristics downwind of the wind farm, by elevating the height of the jet above the wind farm (see Figure 7 and Figure 8),



3. The height of the low-level jet significantly impacts wind farm wake recovery downwind, with higher low-level jet heights resulting in faster wake recovery (see Figure 7 and Figure 8),

4. Negative wind shear during non-LLJ cases show wake propagation at 22 RD downwind of a wind farm,

5. Wind farm wake impacts are observed through the δ (see Figure 11),

6. Gravity waves enhances wake recovery and accelerates winds above the low-level jet downwind of a wind farm (see Figure 13), and lastly,

7. Large-eddy simulation-based $\delta_{IBL}$ models perform well given real-world inputs of the atmosphere and turbine.

Finally, we have highlighted several areas of research in this article that still need to be conducted to understand the dynamics of the wind farm influenced atmospheric boundary layer, mentioned below:

1. Additional comparisons between LES models and observations are required to further evaluate the wake recovery processes within a wind farm wake.

2. It is important to model not only the wind turbine rotor layer with high vertical resolution but up to the top of the δ to accurately assess the impacts of wind farms and wake recovery.

3. Recommend long term (1-year +) evaluation of wind farm wake models or parameterizations using long-term

atmospheric flux observations through the boundary layer to assess the true performance of such models.

4. Analysis would be needed to see the spatial impact of wind farms during gravity wave propagation and power production.

5. Research is needed with large-eddy simulations or numerical weather prediction models with real-world forcing to assess the recovery and growth of $\delta_{IBL}$ at an operational wind farm.

Future work will focus on developing a methodology to classify gravity waves in the region and assess the power production impacts in the presence of gravity waves at the King Plains wind farm site. In addition, comparison of observations and models will be conducted to refine wind farm parameterization schemes within numerical weather models.

**Codes and data statement**

All the data is publicly available on the ARM Discovery webpage (https://adc.arm.gov/discovery/#/) or on the Wind Data Hub (https://www.a2e.energy.gov). The lidar data DOI at site H is 10.21947/2283040 and site A2 is 10.5439/1890922, sonic anemometer data DOI is 10.21947/1899850, finally, ceilometer data at site A1 DOI is 10.21947/2221789.



**Author contribution**

RK and RN designed the experiments. RK developed the code and performed the analysis. RK prepared the manuscript with contributions from all co-authors. CK and PM acquired funding for the analysis. SL, MP, NH, DC, DF, NB assisted in collecting data from various sensors and supported with data analysis.

**Competing interests**

The authors declare that they have no conflict of interest.

**Acknowledgements**

The authors would like to thank the ARM infrastructure for making the data publicly available and to all the mentors for maintaining and processing the instrument data.  The authors would also like to thank the Wind Data Hub team for making the AWAKEN field campaign data publicly available. PNNL is operated for DOE by the Battelle Memorial Institute under Contract DE-AC05-76RLO1830. This work was authored in part by the National Renewable Energy Laboratory, operated
by Alliance for Sustainable Energy, LLC, for the U.S. Department of Energy (DOE) under Contract No. DE-AC36-08GO28308. Funding provided by the U.S. Department of Energy Office of Energy Efficiency and Renewable Energy Wind Energy Technologies Office. The views expressed in the article do not necessarily represent the views of the DOE or the U.S. Government. The U.S. Government retains and the publisher, by accepting the article for publication, acknowledges that the U.S. Government retains a nonexclusive, paid-up, irrevocable, worldwide license to publish or reproduce the
published form of this work, or allow others to do so, for U.S. Government purposes.

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
