# Peer review of "Observations of wind farm wake recovery at an operating wind farm"

_Wind Energy Science, 2024_

## Referee Comment (RC1)

Review of *"Observations of wind farm wake recovery at an operating wind farm"* by Krishnamurthy, R., Newsom, R., Kaul, C., Letizia, S., Pekour, M., Hamilton, N., Chand, D., Flynn, D. M., Bodini, N., and Moriarty, P.

The provided manuscript thoroughly analyses the vertical profiles of the vertical momentum flux and vertical wind speed within a wake induced by a large wind farm in the US Great Plains. In their paper, the authors distinguish between several meteorological parameters, including atmospheric stability, boundary layer height, presence of LLJ events and extreme veer and shear occurrences. Further, the authors provide an exemplary extreme case with a very high downward flux in the wake induced by the presence of a gravity wave. The results show a clear dependence of vertical momentum flux and wind speed deficit on the prevailing atmospheric stability regime, as well as on the presence of extreme events, such as LLJs and in one particular case a gravity wave. Further, observations suggest, that the wind farm's effects are present throughout the entire atmospheric boundary layer, even far above the rotor plane. Thus, the manuscript addresses internationally relevant questions of importance for the scientific community within the scope of the journal.

From my point of view, the language used in the presented manuscript is very nice and the writing style is easy to follow. The chosen title is concise and represents the content of the paper quite well. The authors provide a very thorough and informative literature overview and separate their work from previous research. However, the reference list needs to be checked again as some of the references from the text are missing in the bibliography (e.g. Stevens, 2016 and Parson et al. 2019, Rottman and Simpson, 1989, Draxl et al. 2019).

Within the introduction of the paper, the objective statement is formulated very vague. Instead, I would suggest that the analysis of the wake properties is directly included (cf. comment #7).

The paper's general structure, as well as the presentation of the results, are not reader-friendly. I would suggest reorganizing the paper and first presenting the measurements carried out and elaborating on the data post-processing methodology afterwards. Also, the used measurement devices including the used time frames should be presented more concisely. Further, within the results section, objective description of the results and subjective interpretation a not always distinguishable, which can lead to confusion. Further, some of the Sections provided in the manuscript don't add to the main part of the story and may be moved to an appendix. Further, the main story of the paper could be presented more concisely by adding some of the Sections into an appendix (cf. comment #2).

Also, I think adding some further analysis about the impact of the ABL depth and LLJ characteristics on the observed wake properties would greatly benefit this paper. However, as the results are very original (i.e. observations of momentum flux in the wake of a wind farm and their distinction between the different meteorological circumstances) and interesting for the scientific community, I would like to see an improved version of this manuscript published in the future.

Considering this and the major comments presented in the following, I would recommend the manuscript for a major review.

*General comments:*

1. The structure of the paper makes it hard to grasp many of the underlying principles easily. To gain a thorough understanding of the topic and the results, multiple reads were necessary, with a

lot of jumping in between Sections, to fully understand the whole picture. I think, following the IMRaD structure (Introduction, Methods, Results and Discussion) would benefit the overall presentation of this paper.

    1.1. In my opinion, the presentation of the measurement campaign (Section 4) should be positioned earlier (before Section 2). I think – as this part is really well written and can easily be followed - it would benefit the understanding of the paper and prevent some of the doublings occurring within the paper. All in all, a more concise and concentrated introduction of the measurement locations, devices, scan parameters etc. would be very helpful.

    1.2. Then in Section 3, the flux estimations would follow, as these are the post-processing methods carried out on the collected data.

2. Some of the presented sections are – although very interesting to read – not contributing to the main storyline of the presented paper and should thus be either cut entirely or moved to the appendix.

    2.1. Section 2 (Mathematical preliminaries), may be moved to the appendix as it is beneficial information, but not strictly necessary to follow and understand the general story of the paper.

    2.2. The same is true for Section 6 (Internal Boundary Layer height). The analysis carried out here does not contribute to the objective statement in the introduction of the paper and thus may only be considered additional information and moved to the appendix. Also, some of the information is doubled in Section 2 and Section 5.4 so it may also be integrated into one of these sections to make the paper more concise.

3. The connection between the momentum flux and wake recovery could be worked out in a little more detail.

4. Often (e.g. L.7, L. 70, L.335, L.343) the authors talk about the momentum flux within a wind farm. However, as this is not really what was measured, I suggest aligning with the rest of the formulations saying "downstream", "surrounding" or "within the wind farm wake" …

5. Some variables are introduced in a slightly confusing way. For the atmospheric boundary layer, $\delta$ is introduced, whereas $\delta_{IBL}$ refers to the actual height of the internal boundary layer. Here, the naming of the variables should be consistent to avoid confusion.

6. In section 3.2 a correlation between the different flux estimates is presented. However, very little discussion on the non-negligible scatter between the two estimates is provided in later sections and no explanation on how the observed differences are accounted for in the following analysis is given.

7. In the results section as well as section 3.2, (objective) results and the (subjective) interpretation and discussion of these are very mixed up. I suggest at least introducing a new paragraph when starting the discussion. However, the best case would be to introduce a new section, where a separate discussion of the observations is carried out.

8. In general, abbreviations should be rechecked, as some are either introduced very late or introduced and then not used consistently (e.g. LLJ)

9. Introducing more paragraphs or line breaks would significantly increase the readability of the paper

10. Figures are sometimes labelled (a) and (b),… and sometimes top, left, etc. Here, consistency would be nice. The same is true with the choice of the used lines and markers between all the different profiles.

11. Also, a clear description of how the shown vertical profiles (either via mean or median,…) is missing. It would also be very interesting to see horizontal error bars showing the e.g. standard error of the mean of the profiles to assess the significance of the presented results

12. Mathematical operators should not be written in italic (e.g. sin and cos in L. 156 or log in L. 424) and variables should be in italic (e.g. β in L. 447)
13. The units in the Figure labels are sometimes in round and sometimes in square brackets, please align.
14. Degrees are sometimes represented as °, "deg" or written completely as degrees. Please align.
15. Please check the reference list again. Some of the cited literature is missing. Exemplary are . Stevens, 2016 and Parson et al. 2019, Rottman and Simpson, 1989, Draxl et al. 2019…

*Specific comments:*

1. L. 13: Why are you not mentioning your observations regarding the atmospheric stability here? I think the results are really interesting and worth mentioning in the abstract.
2. L. 20: The provided abstract is more of a teaser of what is to come in the paper. It only provides very limited insight into quantitative results and no qualitative statements.
3. L. 31: Maybe – as you are also talking about offshore wind farms – it makes sense to consider, that during stable stratification offshore wind farms in the German Bight induce wakes are observed from in-situ measurements more than 50km downstream of the wind farm (Platis et al., 2018) and may even cause a detectable decrease in power production for downstream wind farms (Schneemann et al., 2020)
4. L. 35: At first, I was a little confused by the term "rotor layer", maybe a half-sentence explaining what you mean here would be nice
5. L.36f: I do not think, the introduction of the variables u' is necessary here. However, if you choose to do that, please include a quick explanation of the indices and dashes and what they indicate.
6. L. 60: What is meant here by the wake "grows"? Does it grow in space or does the wind speed deficit increase? A little more explanation would be nice.
7. Very vague description of the paper's objective. Instead, I would suggest directly mentioning wake recovery, e.g.: "*In this paper, we investigate the wake recovery of a wind farm, by investigating the momentum balance […]*" (L.74).
8. L. 78-81: I would rather place this part in the conclusion part of the paper.
9. L. 86: As stated before, I would move this part into an appendix, to make the paper more concise. No information necessary for understanding your paper gets lost here.
10. L. 95: Instead of ": The turbulent entrainment of mean kinetic energy", I would rather directly talk about the momentum flux here, or instead provide the prognostic equation for the kinetic energy to highlight the connection between the two.
11. L. 118: I don't fully understand, what is meant by: "$\delta_{IBL}(0)$ is the internal boundary layer height of the wind turbine rotor top". Is it that $\delta_{IBL}(0)$ is equal to the upper tip height? Maybe you could explain this in the text or with a quick equation.
12. L. 144: You could directly introduce the term "Eddy-covariance method" here. This would save you from having to reintroduce how you obtain fluxes from anemometers in L. 180
13. L:170: Here, the measurement technique is introduced. However, if all the different scans along with their locations, devices and time frames would be introduced in one central section (move Section 4 forward) I think this would save a) a lot of space and b) increase the understanding of your measurements.
14. L. 184: Amount of digits used between (a) and (b) are not the same
15. L.186: L as the Monin-Obukhov length has not been introduced up to this point.
16. L.187: Here * is used as a multiplicator, in the figures it is the "convolve" sign. Here, consistency should be achieved.

17. L. 195: Instead of saying "amount of stratification", I would suggest using "strength" or "degree" of stratification.
18. L.203: Here, a cross-correlation between the two different flux estimations could be used to eliminate the difference in measurements due to the spatial displacement (if the devices are oriented in wind directions)
19. L. 209: The discussion regarding the difference between the two flux estimations is rather short. Further, the results are not picked up again in the results or conclusion Sections and thus feel a little lost in the paper.
20. L. 210: In my opinion, this chapter should be moved to the front, as it would support the flux estimation section.
21. L. 211-214: This part is more of an introduction and thus may be placed there.
22. L. 236: Here a reference to the corresponding Equation would be nice.
23. L. 249: The description of the measurement site is very well written and good to follow. However, I would very much appreciate a more in-depth description of the orientation of the lidars and sonic anemometers, to further understand the discrepancies in flux estimations highlighted in Section 3.2. Also, I would suggest to provide a table containing all the different measurement devices, the quantities they measure and the time frame in which they were available. You could also already introduce the fact that only southerly wind sectors were used. All this would then save a lot of space in the following sections and especially the Figure captions.
24. L. 250: The map showing all the different measurement stations is very well done. However, for conciseness, I would suggest zooming in to the relevant part of the area, containing the measurement locations that were actually used. Further, I am missing the location of the ARM SGP central facility to better follow the flux estimation procedure and the Section with the gravity wave measurements.
25. L. 261: A quick table presenting the different stability regimes and borders of $L$ would be helpful
26. L. 262: In my opinion, this chapter could be merged with Section 2, as both deal with the estimation of the (internal) boundary layer height. This would make the paper more concise and the authors would avoid providing similar information at two different stages of the paper.
27. L. 279: How is the statement regarding the momentum flux difference threshold of 1% backed up? Is there literature available?
28. L. 289: In Figure 4, the legend is missing. Also in the title, θ is used as wind direction instead of Φ. Also, as this chapter primarily deals with the IBL height detection, maybe you could provide vertical lines showing the "rotor layer", as well as your mean IBL height.
29. L.308-313: This sounds more like an introduction to me. Or it would also fit in the section describing the methodology to estimate fluxes. However, here I think it distracts a lot from the results.
30. L. 316: The authors refer to a difference in the results due to the diurnal cycle here. I think it would be really interesting if these results were shown in the paper, as they are not present in the referenced Figure.
31. L. 319-320: The authors use expect here a lot. Maybe the sentence could be rephrased to avoid this subsequent use of the word. Further, an explanation on why this is expected is missing.
32. L. 347: In Figure 5 it would be very nice, to visualize the rotor layer (which is stated in the caption, but not present in the Figure). Also, to really emphasize the difference between the situations, which are worked out quite well by the authors, I would suggest that the scaling of the x-axes is kept constant throughout all three subfigures.

33. L. 354: LLJ could be introduced earlier and is not used consistently hereafter.
34. L. 358: I think it would be very interesting, to compare different LLJ definitions, as they are leading to very large differences in analysis. Also, in a recent paper, Hallgren et al. (2023) provide a new concept of LLJ detection using the wind speed shear instead of a fall-off, which seems to be less sensitive to the used measurement device and available height window. I think adding this new definition to your work could make your results more interesting.
35. L.361: Is this stability distribution only for LLJ events or does it include also non-LLJ events? This description is not very clear, also not in the caption of Figure 6.
36. L.368: This result is very interesting and should be shown in a Figure somewhere. From Figure 6, this statement cannot really be verified
37. L.375: Is there any reasoning behind the separation of LLJ events into these height intervals? Some insight into your analysis would be very nice.
38. L. 392: Figure 6 is missing a) and b), there is no legend given. Also, I would like to ask for an explanation of the error bars. Further, I think it would be very interesting – also considering the claims from L. 368 – if instead of the bottom picture, two plots with $Z_{LLJ}$ on the y-axis and $U_{LLJ}$ and $U_{hub}$ on the x-axes respectively were shown. The top picture looks really nice and is very informative. Only a statement about whether all events or only LLJ events are considered in the stability distribution would be nice.
39. L. 400: I think it is more common to use up- and downstream instead of up- and downwind. Also, the authors state, that the LLJ height is modulated when passing through the farm. Here it would be really interesting to see how many LLJ events are recorded in the different core height intervals and maybe also provide another plot showing the LLJ height up- vs downstream of the wind farm.
40. L. 408: I do not think, that Figure 8 provides any significant benefit to the story of the paper. Also, as it does not show the measurements of the analysis by the authors. Instead, I would be really keen to see a quantitative analysis of the indicated LLJ shift upwards during the passing of the farm. Maybe showing the difference in Core height also as distributed over the different stability regimes would be interesting here. However, a similar Figure showing more schematic view of the measurement devices and their location with respect to the wind farm would be very helpful in Section 4.
41. L. 438: Is there a specific reason that the median is used throughout the paper instead of the mean? I think it is valid, if there are large outliers present, but a quick hint on why that is done would be very nice.
42. L. 441-446: Here, you are introducing a new research question, which is usually done in the objective statement within the introduction. Also, I think, this question should have been considered in previous sections, as the presented paper deals with wake recovery throughout all result sections. However, as the authors do not plan to answer this it would be in my opinion more fitting in the Conclusion/Outlook part of the paper. Also, there is a source missing for your presented claim.
43. L. 449: Here, the authors partition their results to events with high and low shear. However, a clear definition of what that means is missing. I would suggest including a description of the distribution of both, α and β, to be able to categorize the division into different veer and shear classes.
44. L. 465-467: I suggest moving this to the introduction or the mathematical preliminaries Section
45. L. 483-488: This sounds more like a conclusion in my opinion.
46. L. 488: Section 5.4, deals not really with the impact of ABL height on wake recovery, but instead concentrates on the extent of the wake within the boundary layer. However, I think that what the title promises is actually a very interesting and important part of what the

entire paper promises. Here, I think it would be very interesting, to different momentum flux profiles for different boundary layer heights and perform the analysis based on that. The way it stands now, I think this section does not provide useful information for the story of the paper.

47. L. 491: This figure is the same as Figure 4 no? What is the added value of providing this plot? As per my previous comment, I really like the idea of this specific analysis and would like to see different momentum flux profiles for different boundary layer heights here. Or, another interesting aspect would be a scatter plot providing the maximum vertical momentum flux vs. the boundary layer height.

48. L. 492: What is the benefit of using the ceilometer measurements over the boundary layer height estimates from the lidar profiles measured at A2 and H? Interesting analysis would also be to have a look at the difference between boundary layer height estimates from the ceilometer and lidars.

49. L. 495: The title suggests an analysis based off of multiple detected gravity waves, when instead only a single event is used for the analysis, thus I would suggest altering the title of this section. This could also be part of the LLJ section, as the following analysis is also based heavily on the LLJ characteristics observed during the event.

50. L. 496-507: This part reads like an Introduction and thus I would suggest moving it there.

51. L. 508-516: This is a nice description of the results, but I would ask to align the date and time representations format with the x-label of Figure 12 and the rest of the paper. Maybe, you could also back up your claim about the observed oscillations being a gravity wave by comparing the observed frequency to the theoretically expected frequency, using e.g. the Brunt-Väisälä frequency.

52. L.526-528: I would consider this comment rather speculative. Can the authors somehow provide a backup for this hypothesis?

53. L. 541: Finally, I think a table analysing the effects of the different effects in comparison with one another would be very helpful to categorise your results (e.g. Comparing average momentum flux deficit at hub height or something similar). This would also add to the discussion in this chapter about what other factors might come into play during this "extreme event" altering the momentum flux and vertical wind profiles.

54. L. 543: The title of Section 6 is very vague and does not represent the content

55. L. 555: Is there any specific reason why only LLJ situations are chosen for this analysis? If yes, I would kindly ask you to provide the reasons to better understand the conclusions followed from that analysis.

56. L. 561: To what standards are the results considered satisfactory?

57. L. 563: In my opinion, this chapter does not add to the main story of the paper. It matches however quite nicely with Section 2 and provides additional material to your paper (esp. Section 5.4). Thus, I would suggest moving it to the appendix. Further, I think it would be very interesting to dig more into the cases when the model is over- and underestimating and to see whether certain patterns can be observed here.

58. L.574: "can" seems a little too strong, as this is not always the case. Instead, I would suggest a "may" here.

59. L. 577-578: This is a very interesting finding. However, it is not really shown in the paper beforehand. As per my previous Comment, I would really like to see this analysis being carried out more thoroughly.

60. L. 583: The first conclusion is not really novel and has been observed before.

61.  L. 588: For this conclusion a categorization on whether this is rather long or short is missing.

62. L. 592: This conclusion is a little bit too generalized. What you show in your paper is purely based on LLJ situations and also there is not really the benchmark defined on what "well" is referring to.
63. L. 595: Maybe, you could explain why this point is important a little. Where are the benefits?
64. L. 599: Here, you could specify the connection to your paper a little better.

*Technical Corrections:*

65. L. 53: Per my understanding it should read "mean winds *within* the ABL"
66. L. 61: I think you are missing a "speed" after wind here.
67. *L. 64: I could not find Stevens (2016) in the reference list, please add that reference (also L. 272 and other such as Parson et al., 2019)*
68. *L. 64: After the citation, some fill word is needed to complete the sentence*
69. *L. 109 & 110: To make the dimensions work, it should be <$u_{z\_h}$> or not? As $c_{ft}$ is dimensionless, you can't multiply speed by height, or else the dimensions would be different as for the first terms with $u^2$ in them.*
70. *L. 112: Usually, the von Kármán (with accent over the a's) is written as κ not k (also in L. 197)*
71. *L. 128: The comma between "lidars" and "and" is not necessary*
72. *L. 138: You are missing the parenthesis around "2020" for the citation*
73. *L. 150: The "R" is missing on the right side of the equation (u(R), v(R), w(R))*
74. *L. 155: Using square brackets here is quite confusing, as one line before, they are used to indicate that the variables are arranged as a vector. Maybe you could just use double round brackets here, to avoid this confusion*
75. *L. 158: <> as the temporal average has already been introduced before*
76. *L. 165: Φ is missing in Eq 5. Maybe small and capital letters are mixed up here.*
77. *L. 303: I think you are referring to Figure 3b here.*
78. *L. 313: Here you are talking about a wind plant, whereas in the rest of the paper you refer to it as wind farm*
79. *L. 314: I would suggest ending the sentence before "Therefore" and starting a new one to improve readability.*
80. *L. 329: I think it should read "convectional" not "conventional" updraft*
81. *L. 400: In the legend, it states that the LLJ core is situated between 100 m and 250 m, I think it is 127 m, no? (cf. L. 374)*
82. L. 424: I think there is one log too much everywhere in this equation. It should read $\log(U(z)) = \log(U(H)) + \alpha \log(z/H)$. Also as log is an operator it should not be written in italic.
83. L. 575: In my understanding, you are referring to Figure 7 or 8, not 14, correct?

*Literature*

Hallgren, C., Aird, J. A., Ivanell, S., Körnich, H., Barthelmie, R. J., Pryor, S. C., and Sahlée, E.: Brief communication: On the definition of the low-level jet, Wind Energ. Sci., 8, 1651–1658, https://doi.org/10.5194/wes-8-1651-2023, 2023.

Platis, A., Siedersleben, S. K., Bange, J., Lampert, A., Bärfuss, K., Hankers, R., Cañadillas, B., Foreman, R., Schulz-Stellenfleth, J., Djath, B., Neumann, T., & Emeis, S. (2018). First in situ evidence of wakes in the far field behind offshore wind farms. *ScIentIfIc REPORtS |*, *8*, 2163. https://doi.org/10.1038/s41598-018-20389-y

Schneemann, J., Rott, A., Dörenkämper, M., Steinfeld, G., and Kühn, M.: Cluster wakes impact on a far-distant offshore wind farm's power, Wind Energ. Sci., 5, 29–49, https://doi.org/10.5194/wes-5-29-2020, 2020.

---

## Referee Comment (RC2)

**Review of the manuscript**

Observations of wind farm wake recovery at an operating wind farm

authored by Raghavendra Krishnamurthy , Rob K. Newsom , Colleen M. Kaul , Stefano Letizia , Mikhail Pekour , Nicholas Hamilton, Duli Chand , Donna Flynn , Nicola Bodini , Patrick Moriarty

**General comments**

The manuscript is about the analysis of an interesting observation data set for vertical momentum fluxes upstream and downstream of wind farms. The study gives insight into the dynamics of wakes depending on the background atmospheric conditions. The study is of high relevance for the validation and improvement of numerical models required to optimize the design and operation of wind farms. The manuscript is well written and understandable. We recommend publication with minor revisions. The identified deficits are mainly related to the theoretical background, which should be explained a little bit more carefully, as well as the spectrum of citations, which could be a little bit broader in some places.

**Specific Comments**

Page 2, Line 49: Maybe one should better say "Todays wind turbines operate …"

Page 2, Line 49: Maybe one could add that in some cases the boundary layer is not even that thick.

Page 2, Line 60: "As wakes grow …" Please be more specific. Do you mean growth in the lateral vertical extend?

Page 2: Please add some brief info about satellite radar wake measurements for offshore wind farms, e.g.

> B. Djath, J. Schulz-Stellenfleth, and B. Canadillas, "Impact of atmospheric stability on X-band and C-band Synthetic Aperture Radar imagery of offshore windpark wakes," *Journal of sustainable and renewable Energy*, vol. 10, no. 4, 2018, doi: 10.1063/1.5020437.

in the intro paragraph mentioning different observation systems. The above publication also points out the importance of a better understanding of vertical momentum fluxes for the interpretation of SAR observations.

Please also mention airborne campaigns, e.g.

> A. Lampert *et al.*, "In situ airborne measurements of atmospheric and sea surface parameters related to offshore wind parks in the German Bight," *Earth System Science Data*, vol. 12, no. 2, pp. 935–946, 2020, doi: 10.5194/essd-12-935-2020.

which also provided info about vertical momentum fluxes.

Page 3, Line 88:  "… change in surface roughness .."

I think this statement is based on a simplified view of the real processes, which is perfectly fine, but this should be stated somehow.  Please cite

> P. Taylor, "On wind and shear stress profiles above a change in surface roughness," *Quarterly Journal of the Royal Meteorological Society*, vol. 95, no. 403, pp. 77–91, 1969.

in this context too.

Page 3, Line 91:  " …growth with downstream distance …"

But it will not grow forever (?)

Page 4, Line 96:  "During stable …"

Did you mean "unstable"  ?

Page 4, Line 103:  Please explain the meaning of the function $F_1$ more carefully (Buckingham Pi theorem, I guess)

Page 4:  I was a little bit confused, because the roughness length $z0$ of the surface without wind farms and the stability seems to be irrelevant in this discussion (?), see e.g.

> S. Emeis, "A simple analytical wind park model considering atmospheric stability," *Wind Energy*, vol. 13, no. 5, pp. 459–469, 2010, doi: 10.1002/we.367].

Please explain this part a little bit more carefully.

Page 5, Line 140:  Please use a different notation for "v", e.g. $v\_\perp$, here. Is it so obvious that $<w>=0$, e.g. in cases with convective cells?

Page 5:  I think a figure explaining the geometry would be helpful.

Figure 1a: The $R^2$ value is hard to believe. I think the reason is that there are so many points on top of each other. Please use a density plot, i.e. 2D histogram. Please indicate in the caption that different axis scaling is used in a) and b).

Page 7, eq. 7: Maybe I missed it somehow, but how did you measure the vertical heat flux?

Page 12, Line 274: "… median streamwise momentum …"
I did not fully understand which upstream/downstream distances the curves in Figure 4 correspond to.

Page 14, Line 303: "… in Figure 4b …"  Did you mean Figure 3 ?

Page 14, Line 324. "Larger …"  please correct sentence.

I think it would be good to learn more about the wind speed profiles upstream to see where we see the largest vertical gradients and where mixing can increase vertical momentum fluxes most effectively.

Page 22: In the context of the discussion about good definitions of wake length one should also mention that it is sometimes not trivial to distinguish wakes from variations in the background wind field, e.g.

B. Djath and J. Schulz-Stellenfleth, "Wind speed deficits downstream offshore wind parks - A new automised estimation technique based on satellite synthetic aperture radar data," *Meteorologische Zeitschrift*, vol. 28, no. 6, pp. 499–515, 2019, doi: 10.1127/metz/2019/0992.

Page 30, line 550: " … upwind surface roughness ($z_{0,hi}$) …"
I'm confused, because I thought $z_{0,hi}$ is the "…roughness due to the presence of a windfarm …" (page 4, line 119)

---

## Author Comment (AC1)

Review of the manuscript Observations of wind farm wake recovery at an operating wind farm authored by Raghavendra Krishnamurthy, Rob K. Newsom , Colleen M. Kaul , Stefano Letizia , Mikhail Pekour , Nicholas Hamilton, Duli Chand , Donna Flynn , Nicola Bodini , Patrick Moriarty

General comments

The manuscript is about the analysis of an interesting observation data set for vertical momentum fluxes upstream and downstream of wind farms. The study gives insight into the dynamics of wakes depending on the background atmospheric conditions. The study is of high relevance for the validation and improvement of numerical models required to optimize the design and operation of wind farms. The manuscript is well written and understandable. We recommend publication with minor revisions. The identified deficits are mainly related to the theoretical background, which should be explained a little bit more carefully, as well as the spectrum of citations, which could be a little bit broader in some places.

We thank the reviewers for their thorough and thoughtful assessment of the article. In the updated manuscript, we have addressed most of the reviewers concerns and provided justification or clarification for others. Our point-by-point responses may be found below in blue font.

Specific Comments

Page 2, Line 49:   Maybe one should better say "Todays wind turbines operate …"

We have updated the manuscript to reflect this statement.

Page 2, Line 49: Maybe one could add that in some cases the boundary layer is not even that thick.

Thanks for the comment.  We agree and have updated the manuscript and stated: "*and in offshore or stable atmospheric conditions the ABL is lower than 300 m (Shaw et al., 2022).*

Page 2, Line 60:  "As wakes grow …"  Please be more specific. Do you mean growth in the lateral vertical extend?

We mean laterally and have made this clear in the updated manuscript.

Page 2: Please add some brief info about satellite radar wake measurements for offshore wind farms, e.g. B. Djath, J. Schulz-Stellenfleth, and B. Canadillas, "Impact of atmospheric stability on X-band and C-band Synthetic Aperture Radar imagery of offshore windpark wakes," *Journal of sustainable and renewable Energy*, vol. 10, no. 4, 2018, doi: 10.1063/1.5020437. in the intro paragraph mentioning different observation systems. The above publication also points out the importance of a better understanding of vertical momentum fluxes for the interpretation of SAR observations.

Please also mention airborne campaigns, e.g. A. Lampert *et al.*, "In situ airborne measurements of atmospheric and sea surface parameters related to offshore wind parks in the German Bight," *Earth System Science Data*, vol. 12, no. 2, pp. 935–946, 2020, doi: 10.5194/essd-12-935-2020. which also provided info about vertical momentum fluxes.

Thank you for alerting us to these two references.  We have now added the above references to the updated manuscript.

Page 3, Line 88:  "… change in surface roughness .."

I think this statement is based on a simplified view of the real processes, which is perfectly fine, but this should be stated somehow.  Please cite P. Taylor, "On wind and shear stress profiles above a change in surface roughness," *Quarterly Journal of the Royal Meteorological Society*, vol. 95, no. 403, pp. 77–91, 1969. in this context too.

We agree and this reference has been added to the updated manuscript.

Page 3, Line 91:  " …growth with downstream distance …" But it

will not grow forever (?)

We agree it will not grow forever and will be capped by the inversion height or the atmospheric boundary layer depth. We mention that in subsequent sentences below.  Therefore, for the sake of not complicating the sentence structure we have left this statement as it is.

Page 4, Line 96:  "During stable …" Did

you mean "unstable"  ?

Perhaps there has been a misunderstanding, we do mean during stable conditions the internal boundary layer height grows to the atmospheric boundary layer height within a short distance, since the atmospheric boundary layer height is shallower, and the wakes are longer.

Page 4, Line 103:  Please explain the meaning of the function $F_1$ more carefully (Buckingham Pi theorem, I guess)

Yes, F is an unknown function and this has been mentioned in the updated manuscript.

Page 4:  I was a little bit confused, because the roughness length z0 of the surface without wind farms and the stability seems to be irrelevant in this discussion (?), see e.g.

S. Emeis, "A simple analytical wind park model considering atmospheric stability," *Wind Energy*, vol. 13, no. 5, pp. 459–469, 2010, doi: 10.1002/we.367|.

Please explain this part a little bit more carefully.

We agree with the reviewer, but these estimates are based on a single column model (Calaf et al., 2010, Stevens, 2016).  One of the authors, Krishnamurthy et al., 2022, has developed an IBL relationship as a

function of atmospheric stability for canonical boundary layers but they are currently not formulated to account for the wind turbine dynamics. This is something that the authors plan to work as a part of future research.

Page 5, Line 140: Please use a different notation for "v", e.g. v_\perp, here. Is it so obvious that <w>=0, e.g. in cases with convective cells?

Since this relates to sonic data post-processing, the 2-axis rotation ensures that the <w> = 0. We would recommend the reviewer to refer Wilczak et al., 2001 for additional information and techniques.

Page 5: I think a figure explaining the geometry would be helpful.
Since this entire section was moved to the Appendix, after reorganization and in the interest of reducing the size of the manuscript, we have referred to the article (Sathe et al., 2015), which provides a geometry used in this manuscript. We have provided a sample image for the scan pattern for the reviewer's benefit.

[Figure]

Figure. Schematic diagrams of Velocity Azimuth Display scan. LiDAR is placed at the origin of the Cartesian coordinate system.

Figure 1a: The $R^2$ value is hard to believe. I think the reason is that there are so many points on top of each other. Please use a density plot, i.e. 2D histogram. Please indicate in the caption that different axis scaling is used in a) and b).

Thank you for the comment. We have mentioned the x-axis scaling difference. Yes, the R2 value is high due to small distribution of observations of momentum flux.

Page 7, eq. 7: Maybe I missed it somehow, but how did you measure the vertical heat flux?

The kinematic heat flux is an estimate from the sonic anemometer. We mention this in the updated

manuscript.

Page 12, Line 274: "… median streamwise momentum …"
  I did not fully understand which upstream/downstream distances the curves in Figure 4 correspond to.

Thank you for the comment. We have now removed this figure, as its repetitive and agree that it did not add much to the manuscript. For other figures, we have provided clear labeling, so hope there is no confusion in the updated manuscript.

Page 14, Line 303: "… in Figure 4b …"   Did you mean Figure 3 ?

Thanks for the typo, this has been fixed.

  Page 14, Line 324. "Larger …"   please correct sentence.

  I think it would be good to learn more about the wind speed profiles upstream to see where we see the largest vertical gradients and where mixing can increase vertical momentum fluxes most effectively.

We agree and have mentioned in our previous statement to "in stable atmospheric conditions, due to large (positive) wind shear, the momentum flux must be negative to create downwind turbulence." The authors have shown many instances where this statement is true in the manuscript.

  Page 22: In the context of the discussion about good definitions of wake length one should also mention that it is sometimes not trivial to distinguish wakes from variations in the background wind field, e.g.

        B. Djath and J. Schulz-Stellenfleth, "Wind speed deficits downstream offshore wind parks - A new automised estimation technique based on satellite synthetic aperture radar data," *Meteorologische Zeitschrift*, vol. 28, no. 6, pp. 499–515, 2019, doi: 10.1127/metz/2019/0992.

Thank you for alerting us to this very interesting paper. We have added it to our references in the updated manuscript.

  Page 30, line 550: " … upwind surface roughness ($z_{0,hi}$) …"
  I'm confused, because I thought $z_{0,hi}$ is the "…roughness due to the presence of a windfarm …" (page 4, line 119)
The reviewer is correct, this typo has been corrected in the updated manuscript.

---

## Author Comment (AC2)

Review of *"Observations of wind farm wake recovery at an operating wind farm"* by Krishnamurthy, R., Newsom, R., Kaul, C., Letizia, S., Pekour, M., Hamilton, N., Chand, D., Flynn, D. M., Bodini, N., and Moriarty, P.

The provided manuscript thoroughly analyses the vertical profiles of the vertical momentum flux and vertical wind speed within a wake induced by a large wind farm in the US Great Plains. In their paper, the authors distinguish between several meteorological parameters, including atmospheric stability, boundary layer height, presence of LLJ events and extreme veer and shear occurrences. Further, the authors provide an exemplary extreme case with a very high downward flux in the wake induced by the presence of a gravity wave. The results show a clear dependence of vertical momentum flux and wind speed deficit on the prevailing atmospheric stability regime, as well as on the presence of extreme events, such as LLJs and in one particular case a gravity wave. Further, observations suggest, that the wind farm's effects are present throughout the entire atmospheric boundary layer, even far above the rotor plane. Thus, the manuscript addresses internationally relevant questions of importance for the scientific community within the scope of the journal.

From my point of view, the language used in the presented manuscript is very nice and the writing style is easy to follow. The chosen title is concise and represents the content of the paper quite well. The authors provide a very thorough and informative literature overview and separate their work from previous research. However, the reference list needs to be checked again as some of the references from the text are missing in the bibliography (e.g. Stevens, 2016 and Parson et al. 2019, Rottman and Simpson, 1989, Draxl et al. 2019).

Within the introduction of the paper, the objective statement is formulated very vague. Instead, I would suggest that the analysis of the wake properties is directly included (cf. comment #7).

The paper's general structure, as well as the presentation of the results, are not reader-friendly. I would suggest reorganizing the paper and first presenting the measurements carried out and elaborating on the data post-processing methodology afterwards. Also, the used measurement devices including the used time frames should be presented more concisely. Further, within the results section, objective description of the results and subjective interpretation a not always distinguishable, which can lead to confusion. Further, some of the Sections provided in the manuscript don't add to the main part of the story and may be moved to an appendix.  Further, the main story of the paper could be presented more concisely by adding some of the Sections into an appendix (cf. comment #2).

Also, I think adding some further analysis about the impact of the ABL depth and LLJ characteristics on the observed wake properties would greatly benefit this paper. However, as the results are very original (i.e. observations of momentum flux in the wake of a wind farm and their distinction between the different meteorological circumstances) and interesting for the scientific community, I would like to see an improved version of this manuscript published in the future.

Considering this and the major comments presented in the following, I would recommend the manuscript for a major review.

We thank the reviewers for their thorough and thoughtful assessment of the article.  In the updated manuscript, we have addressed most of the reviewers concerns and provided justification or clarification for others. Our point-by-point responses may be found below in blue font.

*General comments:*

1. The structure of the paper makes it hard to grasp many of the underlying principles easily. To gain a thorough understanding of the topic and the results, multiple reads were necessary, with a

lot of jumping in between Sections, to fully understand the whole picture. I think, following the IMRaD structure (Introduction, Methods, Results and Discussion) would benefit the overall presentation of this paper.

   1.1. In my opinion, the presentation of the measurement campaign (Section 4) should be positioned earlier (before Section 2). I think – as this part is really well written and can easily be followed - it would benefit the understanding of the paper and prevent some of the doublings occurring within the paper. All in all, a more concise and concentrated introduction of the measurement locations, devices, scan parameters etc. would be very helpful.

We appreciate the suggestion.  We have made some changes to the structure of the manuscript accounting for recommendations made by the reviewer.

   1.2. Then in Section 3, the flux estimations would follow, as these are the post-processing methods carried out on the collected data.

We appreciate the suggestion.  We have made some changes to the structure of the manuscript as mentioned above.

2. Some of the presented sections are – although very interesting to read – not contributing to the main storyline of the presented paper and should thus be either cut entirely or moved to the appendix.

   2.1. Section 2 (Mathematical preliminaries), may be moved to the appendix as it is beneficial information, but not strictly necessary to follow and understand the general story of the paper.

We agree and have made some changes to the structure of the manuscript.

   2.2. The same is true for Section 6 (Internal Boundary Layer height). The analysis carried out here does not contribute to the objective statement in the introduction of the paper and thus may only be considered additional information and moved to the appendix. Also, some of the information is doubled in Section 2 and Section 5.4 so it may also be integrated into one of these sections to make the paper more concise.

We have integrated the text as requested, but we believe the Internal Boundary Layer height and momentum recovery of wind farms are linked and deserves to be in the new Section 5. Especially during cases of low-level jets, the momentum recovery and internal boundary layer are a function of the LLJ height (as shown in other sections).  Moreover, this section gives us a sense of uncertainty of LES based estimates of IBLs, which is important for advancing future work in this area.

3. The connection between the momentum flux and wake recovery could be worked out in a little more detail.

While addressing the manuscript updates, we believe this aspect has become clearer, thanks to the reviewer's comments.

4. Often (e.g. L.7, L. 70, L.335, L.343) the authors talk about the momentum flux within a wind farm. However, as this is not really what was measured, I suggest aligning with the rest of the formulations saying "downstream", "surrounding" or "within the wind farm wake" …

Thanks for the comment. We have changed it to "surrounding" where appropriate.

5. Some variables are introduced in a slightly confusing way. For the atmospheric boundary layer, $\delta$ is introduced, whereas $\delta_{IBL}$ refers to the actual height of the internal boundary layer. Here, the naming of the variables should be consistent to avoid confusion.

Thanks for the comment. We have tried to stay consistent in the updated manuscript.

6. In section 3.2 a correlation between the different flux estimates is presented. However, very little discussion on the non-negligible scatter between the two estimates is provided in later sections and no explanation on how the observed differences are accounted for in the following analysis is given.

We do mention some of the reasons for the scatter and have placed that paragraphs from manuscript (in quotes) below for the reviewer. This section has now been moved to the Appendix A.2 upon reviewers' recommendation above. The comparison was done near a 60-m flux tower near the ARM SGP site C1 and a near-by Doppler lidar.

"During stable atmospheric conditions, given the amount of stratification within the lidar probe volume, the lidar could be measuring very different atmospheric conditions compared to a sonic anemometer." and

"The coefficient of determination is observed to reduce during stable conditions to ~63%, although the wind speeds are observed to correlate well under all conditions. The transfer of momentum is lowest in stable atmospheric conditions and therefore smaller momentum flux estimates are observed. From a purely statistical standpoint, the smaller magnitude of the fluxes also contributes to reducing the coefficient of determination, since under these conditions the contribution of instrumental and statistical noise to the physical variability of relatively larger. The scatter between lidar and sonic measurements are primarily due to (a) 15 m vertical and ~250 m horizontal separations between sonic anemometer and lidar measurements, (b) low temporal sampling of the lidar measurements, and (c) spatial averaging of the lidar pulse (range-gate = 30 m). These effects amplify during stable atmospheric conditions and result in larger scatter between measurements. Previous observations of momentum flux from profiling Doppler lidars have shown a similar accuracy when compared to sonic anemometers at various heights above ground level (Mann et al., 2010)."

With regards to the current analysis, the lowest observation level (4 m) in all of the momentum flux profiles shown in the article are from the co-located sonic anemometers at Site A2 (Site H didn't have a sonic anemometer for the dates analyzed in this paper). As its observed, the median flux observations are consistent with lidar observations. In the updated manuscript, we have added some error bars to show the spread in some of these atmospheric conditions. For the above reasons mentioned, we have not really applied any specific correction to the lidar observations as we believe the spread was mostly due to different volume sampling and temporal sampling of the two datasets.

7. In the results section as well as section 3.2, (objective) results and the (subjective) interpretation and discussion of these are very mixed up. I suggest at least introducing a new paragraph when starting the discussion. However, the best case would be to introduce a new section, where a separate discussion of the observations is carried out.

Thanks for the comment. We have attempted to add a new paragraphs to make these sections more readable.

8. In general, abbreviations should be rechecked, as some are either introduced very late or introduced and then not used consistently (e.g. LLJ)

Thanks for the comment. It is currently consistent in the manuscript and defined in the introduction.

9. Introducing more paragraphs or line breaks would significantly increase the readability of the paper

We have added attempted to add new paragraphs where appropriate to make these sections more readable.

10. Figures are sometimes labelled (a) and (b),… and sometimes top, left, etc. Here, consistency would be nice. The same is true with the choice of the used lines and markers between all the different profiles.

These have been made consistent in the updated manuscript.

11. Also, a clear description of how the shown vertical profiles (either via mean or median,…) is missing. It would also be very interesting to see horizontal error bars showing the e.g. standard error of the mean of the profiles to assess the significance of the presented results

Thanks for the comment. We initially felt adding error bars generally muddy up the figure hence did not include them, but we agree that it is importance to assess the significance of the presented results. We have added the standard deviation of the observations for atleast one of the figures showing impact of wake recovery during various atmospheric stability conditions. All observations shown are the median profiles, which is our preferred way to show observations as sometimes the filtering of lidar data can get finicky at subjective. Therefore, they can skew the observations.

12. Mathematical operators should not be written in italic (e.g. sin and cos in L. 156 or log in L. 424) and variables should be in italic (e.g. β in L. 447)

These have been made consistent in the updated manuscript.

13. The units in the Figure labels are sometimes in round and sometimes in square brackets, please align.

These have been made consistent in the updated manuscript.

14. Degrees are sometimes represented as °, "deg" or written completely as degrees. Please align.

These have been made consistent in the updated manuscript.

15. Please check the reference list again. Some of the cited literature is missing. Exemplary are . Stevens, 2016 and Parson et al. 2019, Rottman and Simpson, 1989, Draxl et al. 2019…

These have been made consistent in the updated manuscript. We apologize for this oversight.

*Specific comments:*

1. L. 13: Why are you not mentioning your observations regarding the atmospheric stability here? I think the results are really interesting and worth mentioning in the abstract.

Thanks for the comment. We have now also added the impacts of atmospheric stability within the abstract.

2. L. 20: The provided abstract is more of a teaser of what is to come in the paper. It only provides very limited insight into quantitative results and no qualitative statements.

We have now also added additional quantitative results in the abstract.

3. L. 31: Maybe – as you are also talking about offshore wind farms – it makes sense to consider, that during stable stratification offshore wind farms in the German Bight induce wakes are observed from in-situ measurements more than 50km downstream of the wind farm (Platis et al., 2018) and may even cause a detectable decrease in power production for downstream wind farms (Schneemann et al., 2020)

We agree and these two references have been now added to the updated manuscript.

4. L. 35: At first, I was a little confused by the term "rotor layer", maybe a half-sentence explaining what you mean here would be nice

We have now defined a rotor layer, "the layer from the bottom of the wind turbine blade tip to the top of the blade tip".

5. L.36f: I do not think, the introduction of the variables u' is necessary here. However, if you choose to do that, please include a quick explanation of the indices and dashes and what they indicate.

We agree and have deleted them in the updated manuscript.

6. L. 60: What is meant here by the wake "grows"? Does it grow in space or does the wind speed deficit increase? A little more explanation would be nice.

We have rephrased it to "As wake extent grows laterally downwind of a wind farm…"

7. Very vague description of the paper's objective. Instead, I would suggest directly mentioning wake recovery, e.g.: "*In this paper, we investigate the wake recovery of a wind farm, by investigating the momentum balance […]*" (L.74).

Thank you for the suggestion and we agree. We have adapted this phrasing in the updated manuscript.

8. L. 78-81: I would rather place this part in the conclusion part of the paper.

We have removed these lines and incorporated parts of it in the conclusion of the paper.

9. L. 86: As stated before, I would move this part into an appendix, to make the paper more concise. No information necessary for understanding your paper gets lost here.

We have integrated the section previously known as "Mathematical Preliminaries" to the Introduction of the paper and moved the section "Flux estimation algorithms and approach" to the appendix. We agree that these are supplemental details and are not truly required for the understanding of the paper.

10. L. 95: Instead of ": The turbulent entrainment of mean kinetic energy", I would rather directly talk about the momentum flux here, or instead provide the prognostic equation for the kinetic energy to highlight the connection between the two.

We directly mention momentum flux here in the updated manuscript.

11. L. 118: I don't fully understand, what is meant by: "$\delta_{IBL}(0)$ is the internal boundary layer height of the wind turbine rotor top". Is it that $\delta_{IBL}(0)$ is equal to the upper tip height? Maybe you could explain this in the text or with a quick equation.

Yes, $\delta IBL(0)$ is the upper tip height. We now mention "…equal to the wind turbine blade upper tip height…".

12. L. 144: You could directly introduce the term "Eddy-covariance method" here. This would save you from having to reintroduce how you obtain fluxes from anemometers in L. 180

This entire section has been moved to the Appendix as mentioned earlier and we have now made the above change to the text in the updated manuscript.

13. L:170: Here, the measurement technique is introduced. However, if all the different scans along with their locations, devices and time frames would be introduced in one central section (move Section 4 forward) I think this would save a) a lot of space and b) increase the understanding of your measurements.

We agree and have moved the previous section 4 to Section 2 in the updated manuscript.

14. L. 184: Amount of digits used between (a) and (b) are not the same

The number of digits for the $R^2$ and linear fit equation in both figures are made the same.

15. L.186: L as the Monin-Obukhov length has not been introduced up to this point.

The definition of the Obukhov length has been moved to Section 2 of the main manuscript.

16. L.187: Here * is used as a multiplicator, in the figures it is the "convolve" sign. Here, consistency should be achieved.

The figures have been updated to not show the convolve sign.

17. L. 195: Instead of saying "amount of stratification", I would suggest using "strength" or "degree" of stratification.

We have changed it to "degree" of stratification.

18. L.203: Here, a cross-correlation between the two different flux estimations could be used to eliminate the difference in measurements due to the spatial displacement (if the devices are oriented in wind directions)

We agree, but our goal was mainly to show the good correlation observed between the two sensors despite the differences. The impact of spatial averaging and lower temporal sampling is expected to have a larger impact (Mann et al., 2010).

19. L. 209: The discussion regarding the difference between the two flux estimations is rather short. Further, the results are not picked up again in the results or conclusion Sections and thus feel a little lost in the paper.

The goal of this section (currently moved to Appendix, thanks to your suggestion above) was mainly to show the accuracy of this technique compared to sonic anemometers at SGP. These results here have no direct bearing on the AWAKEN observations. By moving this to the Appendix, we believe this issue has been resolved in the updated manuscript.

20. L. 210: In my opinion, this chapter should be moved to the front, as it would support the flux estimation section.

We agree and is currently Section 2 of the updated manuscript.

21. L. 211-214: This part is more of an introduction and thus may be placed there.

We believe these details about the Oklahoma region, although introductory material, ties into the location of the AWAKEN campaign (which is not discussed explicitly in the introduction). Moreover, it doesn't really fit anywhere in the introductory material as currently written up in the updated manuscript. To avoid re-writing the introduction just to accommodate these details, we would prefer it in its current location (Section 2 of the updated manuscript).

22. L. 236: Here a reference to the corresponding Equation would be nice.

We have now referenced the corresponding equation.

23. L. 249: The description of the measurement site is very well written and good to follow. However, I would very much appreciate a more in-depth description of the orientation of the lidars and sonic anemometers, to further understand the discrepancies in flux estimations highlighted in Section 3.2. Also, I would suggest to provide a table containing all the different measurement devices, the quantities they measure and the time frame in which they were available. You could also already introduce the fact that only southerly wind sectors were used. All this would then save a lot of space in the following sections and especially the Figure captions.

Thank you for this suggestion. We have added some additional details about the orientation of the lidars and sonic anemometers. And mentioned about the southerly wind sectors being used in the analysis. With regards to measurement devices and quantities they measure, we have just referred to an overview article currently in press within Journal of Renewable and Sustainable Energy.

24. L. 250: The map showing all the different measurement stations is very well done. However, for conciseness, I would suggest zooming in to the relevant part of the area, containing the measurement locations that were actually used. Further, I am missing the location of the ARM SGP central facility to better follow the flux estimation procedure and the Section with the gravity wave measurements.

Noted and an updated zoomed out map is also provided in the updated manuscript.

25. L. 261: A quick table presenting the different stability regimes and borders of $L$ would be helpful

We have added Table 1 now in the updated manuscript, providing details of the various stability regime classifications.

26. L. 262: In my opinion, this chapter could be merged with Section 2, as both deal with the estimation of the (internal) boundary layer height. This would make the paper more concise and the authors would avoid providing similar information at two different stages of the paper.

This section has been moved as discussed earlier.

27. L. 279: How is the statement regarding the momentum flux difference threshold of 1% backed up? Is there literature available?

Unfortunately, there is no literature reference available.  We hypothesized a small threshold would be required to accurately quantify the internal boundary layer height. We have not done extensive sensitivity studies to this threshold and plan to conduct a more thorough analysis in the future.

28. L. 289: In Figure 4, the legend is missing. Also in the title, θ is used as wind direction instead of Φ. Also, as this chapter primarily deals with the IBL height detection, maybe you could provide vertical lines showing the "rotor layer", as well as your mean IBL height.

We have removed this figure from the paper, as this was a redundant figure and with the restructuring it was not a valuable addition anymore.  But appreciate the reviewer's comments in better representation.  We have added rotor layer lines to other similar figures.

29. L.308-313: This sounds more like an introduction to me. Or it would also fit in the section describing the methodology to estimate fluxes. However, here I think it distracts a lot from the results.

We have decided to keep the sentence here, as we feel it adds some value to the discussion.

30. L. 316: The authors refer to a difference in the results due to the diurnal cycle here. I think it would be really interesting if these results were shown in the paper, as they are not present in the referenced Figure.

We really meant that the variability observed during stable (primarily nighttime) and unstable (primarily daytime) conditions, which is shown in the plots.

31. L. 319-320: The authors use expect here a lot. Maybe the sentence could be rephrased to avoid this subsequent use of the word. Further, an explanation on why this is expected is missing.

We have rephrased it to "Under neutral conditions, where shear is less positive and ambient turbulence is higher compared to stable conditions, the momentum flux generated by downwind wind turbines is anticipated to be lower or less persistent.  Consequently, wakes are not expected to travel as far."

32. L. 347: In Figure 5 it would be very nice, to visualize the rotor layer (which is stated in the caption, but not present in the Figure).  Also, to really emphasize the difference between the situations, which are worked out quite well by the authors, I would suggest that the scaling of the x-axes is kept constant throughout all three subfigures.

We apologize for not showing the rotor layer but have updated the figure to show the above.

33. L. 354: LLJ could be introduced earlier and is not used consistently hereafter.

It's been made consistent in the manuscript.

34. L. 358: I think it would be very interesting, to compare different LLJ definitions, as they are leading to very large differences in analysis. Also, in a recent paper, Hallgren et al. (2023) provide a new concept of LLJ detection using the wind speed shear instead of a fall-off, which seems to be less sensitive to the used measurement device and available height window. I think adding this new definition to your work could make your results more interesting.

We completely agree but we feel this is currently out of scope of the paper. We would expect another publication highlighting the impact of different LLJ definitions to be conducted as a part of future analysis.

35. L.361: Is this stability distribution only for LLJ events or does it include also non-LLJ events? This description is not very clear, also not in the caption of Figure 6.

In the text we had mentioned that these are only for LLJ events and that too only southerly wind directions but have revised the statement. We have also added this to the caption.

36. L.368: This result is very interesting and should be shown in a Figure somewhere. From Figure 6, this statement cannot really be verified

We have rephrased this sentence, and it currently reads: "Figure 4b shows LLJ nose wind speed as a function of median $Z_{LLJ}$ per wind speed bin and hub-height wind speed and Figure 4c shows $Z_{LLJ}$ as a function of hub-height wind speed. It is evident that higher the $Z_{LLJ}$, higher the jet nose wind speed and higher the hub-height wind speed."

Please see below the updated figure in the text and associated caption.

[Figure]

Figure 4. (a) Distribution of various atmospheric stability classes (VS – Very Stable, Stable, NNS – Near-Neutral Stable, Neutral, NNUS – Near-Neutral Unstable, US – Unstable, VUS – Very Unstable, as per Sathe et al., 2015) during LLJ events from southerly wind directions and associated $Z_{LLJ}$ per stability class, (b) median LLJ nose wind speed ($U_{LLJnose}$) as a function of $Z_{LLJ}$ and hub-height wind speed ($U_{hub}$) at the upwind site (site A2), and (c) $Z_{LLJ}$ as a function of $U_{hub}$. The error bars indicate one standard deviation. Minimum $Z_{LLJ}$ is 110 m and maximum $Z_{LLJ}$ is 690 m AGL. Measurements only from southerly wind directions, especially from 166 deg to 190 deg, and from 17 March 2023 to 10 September 2023 are considered in this analysis.

37. L.375: Is there any reasoning behind the separation of LLJ events into these height intervals? Some insight into your analysis would be very nice.

We have added some insights as shown below.

"The partitioning was driven by selecting a height near the wind turbine rotor layer (25.5 m to 152.5 m) that could be impacted by the wind turbine, considering the frequency of LLJ events from southerly wind directions (which peaked around 250 m above ground level), and the observed peak in momentum flux during stable conditions, which occurred approximately 250 m above ground level as shown in Figure 3."

38. L. 392: Figure 6 is missing a) and b), there is no legend given. Also, I would like to ask for an explanation of the error bars. Further, I think it would be very interesting – also considering the claims from L. 368 – if instead of the bottom picture, two plots with $Z_{LLJ}$ on the y-axis and $U_{LLJ}$ and $U_{hub}$ on the x-axes respectively were shown. The top picture looks really nice and is very informative. Only a statement about whether all events or only LLJ events are considered in the stability distribution would be nice.

We have modified the figure as requested and others have been addressed based off prior comments.

39. L. 400: I think it is more common to use up- and downstream instead of up- and downwind. Also, the authors state, that the LLJ height is modulated when passing through the farm. Here it would be really interesting to see how many LLJ events are recorded in the different core height intervals and maybe also provide another plot showing the LLJ height up- vs downstream of the wind farm.

Thanks for the comment, we initially had a similar plot but removed it due to concerns on length of the manuscript. We have added the below figure and associated text.

"Figure 7 illustrates the probability distribution of LLJ events at the upwind site (A2) and downwind site (H) of King Plains wind farm. The data reveal that the difference in LLJ height between the upwind and downwind sites is greater below approximately 250 to 300 m but diminishes at higher LLJ heights. Notably, there is a reduced frequency of LLJs observed downwind of the wind farm when LLJs occur below the rotor layer. Future research will focus on further analyzing the effects of LLJs that occur beneath the turbine rotor layer.

[Figure]

Figure 7. Probability distribution of LLJ heights (ZLLJ) upwind (site A2) and downwind (site H) of the King Plains wind farm during southerly wind directions."

40. L. 408: I do not think, that Figure 8 provides any significant benefit to the story of the paper. Also, as it does not show the measurements of the analysis by the authors. Instead, I would be really keen to see a quantitative analysis of the indicated LLJ shift upwards during the passing of the farm. Maybe showing the difference in Core height also as distributed over the different stability regimes would be interesting here. However, a similar Figure showing more schematic view of the measurement devices and their location with respect to the wind farm would be very helpful in Section 4.

We have found that visual schematics significantly enhance the readability of the paper for students and researchers who may not have a high level of expertise in the subject. Therefore, we would like to retain this figure (currently Figure 6 in the updated manuscript) in the manuscript.

41. L. 438: Is there a specific reason that the median is used throughout the paper instead of the mean? I think it is valid, if there are large outliers present, but a quick hint on why that is done would be very nice.

Yes, with observations as it is difficult to completely remove all the outliers, we prefer to show the median instead of the mean. We have mentioned this in Section 2 of the updated manuscript.

42. L. 441-446: Here, you are introducing a new research question, which is usually done in the objective statement within the introduction. Also, I think, this question should have been considered in previous sections, as the presented paper deals with wake recovery throughout all result sections. However, as the authors do not plan to answer this it would be

in my opinion more fitting in the Conclusion/Outlook part of the paper. Also, there is a source missing for your presented claim.

We have removed this sentence from the updated manuscript.

43. L. 449: Here, the authors partition their results to events with high and low shear. However, a clear definition of what that means is missing. I would suggest including a description of the distribution of both, α and β, to be able to categorize the division into different veer and shear classes.

We have mentioned the percentage of times such events were observed in the caption of the respective figures.

44. L. 465-467: I suggest moving this to the introduction or the mathematical preliminaries Section

This section has been moved to the introduction.

45. L. 483-488: This sounds more like a conclusion in my opinion.

This statement was deleted as it was already in the conclusions.

46. L. 488: Section 5.4, deals not really with the impact of ABL height on wake recovery, but instead concentrates on the extent of the wake within the boundary layer. However, I think that what the title promises is actually a very interesting and important part of what the entire paper promises. Here, I think it would be very interesting, to different momentum flux profiles for different boundary layer heights and perform the analysis based on that. The way it stands now, I think this section does not provide useful information for the story of the paper.

We have revised the title to "Extent of Wake Within the ABL." While we agree that this is an important and intriguing topic, it is currently being investigated by other researchers within the project. We will pass this comment along to them.

47. L. 491: This figure is the same as Figure 4 no? What is the added value of providing this plot? As per my previous comment, I really like the idea of this specific analysis and would like to see different momentum flux profiles for different boundary layer heights here. Or, another interesting aspect would be a scatter plot providing the maximum vertical momentum flux vs. the boundary layer height.

Figure 4 of the older manuscript was deleted; hence it is not repetitive. As mentioned in the comment above, we will pass this comment to other researchers working on this topic.

48. L. 492: What is the benefit of using the ceilometer measurements over the boundary layer height estimates from the lidar profiles measured at A2 and H? Interesting analysis would also be to have a look at the difference between boundary layer height estimates from the ceilometer and lidars.

Very interesting question, but the Lidar does not provide boundary layer heights during night-time conditions. As the estimation is based on a vertical velocity variance threshold (Tucker et al., 2009, Krishnamurthy et al., 2022), the variance is near zero during night-time conditions and sometimes beneath the first range-gate of the Doppler lidar.

49. L. 495: The title suggests an analysis based off of multiple detected gravity waves, when instead only a single event is used for the analysis, thus I would suggest altering the title of this section. This could also be part of the LLJ section, as the following analysis is also based heavily on the LLJ characteristics observed during the event.

Agreed, the title was changed to "Impact of a gravity wave case on wake recovery".

50. L. 496-507: This part reads like an Introduction and thus I would suggest moving it there.

Agreed, it has been moved to the introduction in the updated manuscript.

51. L. 508-516: This is a nice description of the results, but I would ask to align the date and time representations format with the x-label of Figure 12 and the rest of the paper. Maybe, you could also back up your claim about the observed oscillations being a gravity wave by comparing the observed frequency to the theoretically expected frequency, using e.g. the Brunt-Väisälä frequency.

We are not sure we understand the reviewer's comment about the time format, if you could kindly rephrase that would help.

We unfortunately do not have all the observations to estimate Brunt-Väisälä frequency during this case study (the thermodynamic profilers were not fully operational and no radiosonde releases were conducted at the same time).

52. L.526-528: I would consider this comment rather speculative. Can the authors somehow provide a backup for this hypothesis?

We have removed this sentence in the updated manuscript.

53. L. 541: Finally, I think a table analysing the effects of the different effects in comparison with one another would be very helpful to categorise your results (e.g. Comparing average momentum flux deficit at hub height or something similar). This would also add to the discussion in this chapter about what other factors might come into play during this "extreme event" altering the momentum flux and vertical wind profiles.

Thank you for your comment. We believe this information may not be very useful to the reader and could be somewhat repetitive of the text above.

54. L. 543: The title of Section 6 is very vague and does not represent the content

We have revised the title to "Comparisons of observed $\delta_{IBL}$ with theoretical estimates".

55. L. 555: Is there any specific reason why only LLJ situations are chosen for this analysis? If yes, I would kindly ask you to provide the reasons to better understand the conclusions followed from that analysis.

We have provided a reasoning in the updated manuscript.

"Since we have higher confidence in $\delta_{IBL}$ during these cases and $\delta$ already represents the top of the LLJ height, this also avoids introducing additional uncertainty from Ceilometer $\delta$ observations."

56. L. 561: To what standards are the results considered satisfactory?

We have removed that statement from the manuscript.

57. L. 563: In my opinion, this chapter does not add to the main story of the paper. It matches however quite nicely with Section 2 and provides additional material to your paper (esp. Section 5.4). Thus, I would suggest moving it to the appendix. Further, I think it would be very interesting to dig more into the cases when the model is over- and underestimating and to see whether certain patterns can be observed here.

We feel this paper provides some insight into how large-eddy simulation model theories scale up with observations of internal boundary layers, which is tightly coupled with wake recovery. Further research is being conducted to extend this analysis in the direction the reviewer is recommending.

58. L.574: "can" seems a little too strong, as this is not always the case. Instead, I would suggest a "may" here.

Agreed, we have changed it to "may".

59. L. 577-578: This is a very interesting finding. However, it is not really shown in the paper beforehand. As per my previous Comment, I would really like to see this analysis being carried out more thoroughly.

As mentioned previously, we will have to defer this as a part of future research within the project.

60. L. 583: The first conclusion is not really novel and has been observed before.

Agreed, its removed.

61. L. 588: For this conclusion a categorization on whether this is rather long or short is missing.

We have added "short".

62. L. 592: This conclusion is a little bit too generalized. What you show in your paper is purely based on LLJ situations and also there is not really the benchmark defined on what "well" is referring to.

This has been rephrased to "Large-eddy simulation-based theoretical $\delta_{IBL}$ models perform well show a large spread given real-world inputs of the atmosphere and turbine."

63. L. 595: Maybe, you could explain why this point is important a little. Where are the benefits?

We have added this sentence: "Since most mesoscale wake model parameterizations are assessed using outputs from LES models."

64. L. 599: Here, you could specify the connection to your paper a little better.

We have modified the text to : "It is important to model not only the wind turbine rotor layer with high vertical resolution but up to the top of the δ to accurately assess the impacts of wind farms and wake recovery (as shown in Figure 9). As it is important to understand the entrainment of winds from the ABL to the wind farm wake boundary layer."

*Technical Corrections:*

65. L. 53: Per my understanding it should read "mean winds *within* the ABL"

We have made the correction.

66. L. 61: I think you are missing a "speed" after wind here.
We have made the correction.

67. L. 64: I could not find Stevens (2016) in the reference list, please add that reference (also L. 272 and other such as Parson et al., 2019)
We have added this reference and a few others noted by the other reviewer.

68. L. 64: After the citation, some fill word is needed to complete the sentence
We have made the correction.

69. L. 109 & 110: To make the dimensions work, it should be $$ or not? As $c_{ft}$ is dimensionless, you can't multiply speed by height, or else the dimensions would be different as for the first terms with $u^2$ in them.
We have made the correction.

70. L. 112: Usually, the von Kármán (with accent over the a's) is written as κ not k (also in L. 197)
We have made the correction.

71. L. 128: The comma between "lidars" and "and" is not necessary
We have made the correction.

72. L. 138: You are missing the parenthesis around "2020" for the citation
We have made the correction.

73. L. 150: The "R" is missing on the right side of the equation (u(R), v(R), w(R))
Thank you. We have made the correction.

74. L. 155: Using square brackets here is quite confusing, as one line before, they are used to indicate that the variables are arranged as a vector. Maybe you could just use double round brackets here, to avoid this confusion
Thank you. We have made the correction.

75. L. 158: <> as the temporal average has already been introduced before *76.* L. 165: Φ is missing in Eq 5. Maybe small and capital letters are mixed up here.
Thank you. We have made the correction.

*77.* L. 303: I think you are referring to Figure 3b here.
We have made the correction. The numbering has although changed in the updated manuscript.

*78.* L. 313: Here you are talking about a wind plant, whereas in the rest of the paper you refer to it as wind farm
Thank you, we have made the correction in the updated manuscript.

*79.* L. 314: I would suggest ending the sentence before "Therefore" and starting a new one to improve readability.

We have ended the sentence and started a new paragraph.

    *80.* L. 329: I think it should read "convectional" not "conventional" updraft

We have made the correction in the updated manuscript.

    *81.* L. 400: In the legend, it states that the LLJ core is situated between 100 m and 250 m, I think it is 127 m, no? (cf. L. 374)

We have made the correction in the updated manuscript.  This was a typo, and should be 110 m AGL.

    *82.* L. 424: I think there is one log too much everywhere in this equation. It should read log(U(z))=log(U(H)) + α log(z/H). Also as log is an operator it should not be written in italic.

Agreed, we have made the correction.

    *83.* L. 575: In my understanding, you are referring to Figure 7 or 8, not 14, correct?

 Yes, we have made the correction.

*Literature*

Hallgren, C., Aird, J. A., Ivanell, S., Körnich, H., Barthelmie, R. J., Pryor, S. C., and Sahlée, E.: Brief communication: On the definition of the low-level jet, Wind Energ. Sci., 8, 1651–1658, https://doi.org/10.5194/wes-8-1651-2023, 2023.

Platis, A., Siedersleben, S. K., Bange, J., Lampert, A., Bärfuss, K., Hankers, R., Cañadillas, B., Foreman, R., Schulz-Stellenfleth, J., Djath, B., Neumann, T., & Emeis, S. (2018). First in situ evidence of wakes in the far field behind offshore wind farms. *ScIentIfIc REPORtS |*, *8*, 2163. https://doi.org/10.1038/s41598-018-20389-y

Schneemann, J., Rott, A., Dörenkämper, M., Steinfeld, G., and Kühn, M.: Cluster wakes impact on a far-distant offshore wind farm's power, Wind Energ. Sci., 5, 29–49, https://doi.org/10.5194/wes-529-2020, 2020.

The last two references have been added to the manuscript.

---

## Referee Report (RR1)

Review of *"Observations of wind farm wake recovery at an operating wind farm"* by Krishnamurthy, R., Newsom, R., Kaul, C., Letizia, S., Pekour, M., Hamilton, N., Chand, D., Flynn, D. M., Bodini, N., and Moriarty, P.

The revised manuscript analyses the vertical profiles of the vertical momentum flux and vertical wind speed within a wake induced by a large wind farm in the US Great Plains. In their paper, the authors distinguish between several meteorological parameters, including atmospheric stability, boundary layer height, presence of LLJ events and extreme veer and shear occurrences. Further, the authors provide an exemplary extreme case with a very high downward flux in the wake, possibly induced by the presence of a gravity wave. The results show a clear dependence of vertical momentum flux and wind speed deficit on the prevailing atmospheric stability regime, as well as on the presence of extreme events, such as LLJs and in one particular case a gravity wave. Further, observations suggest, that the wind farm's effects are present throughout the entire atmospheric boundary layer, even far above the rotor plane. Thus, the manuscript addresses internationally relevant questions of importance for the scientific community within the scope of the journal.

From my point of view, the language used in the presented manuscript is very nice and the writing style is easy to follow. The chosen title is concise and represents the content of the paper quite well. The authors provide a very thorough and informative literature overview and separate their work from previous research.

Within the introduction of the paper, the objective statement is formulated very vague. Instead, I would suggest that the analysis of the wake properties is directly included (cf. comment #7).

The structure of the revised manuscript is now easier to grasp and follows a clear storyline. Sections that do not directly add value to the main objective of the paper are now moved to the appendix and provide valuable additional information on the measurements and post-processing.

Considering this and the comments presented in the following, I would recommend the manuscript for a minor revision.

*General comments:*

1. The authors jump between abbreviations and the written version of LLJ and low-level jet. (e.g. L. 10, 14, 16, 24, 278, etc.)
2. Sometimes, adding a "the" would lead to increased readability, e.g. L.236: "[The] larger the vertical momentum flux, [the] faster the wake [...]" or L. 241-242: "the impact of conventional updrafts or downdrafts on [the] propagation of wakes"
3. In Fig. 3 error bars are given for the median profiles, while they are missing in Fig. 5 and following. Is there any specific reasoning behind this?

*Specific comments:*

4. L. 28: You only mention mesoscale simulations here, but e.g. in Schneemann et al. 2020 the authors observed them with scanning offshore lidars.
5. L. 53: The formulation "ABL is lower than 300m" suggests that this is always the case. However, as per my knowledge, even in stable conditions boundary layer height can exceed 300m (e.g. Peng et al. 2023).

6. L. 209: Here, the authors claim that "sufficient" data is available. It would be helpful to know, how much (e.g. in hours or No. of measurements) that is.
7. L. 229: The authors claim, that larger momentum flux deficits for near surface areas are observed for unstable and neutral conditions. However, I would argue that during stable conditions, based on the provided figures, the momentum flux deficit is larger than for neutral conditions and also compared to unstable conditions. At larger heights it then seems as if momentum flux deficits are larger during unstable conditions. Maybe a clearer picture containing directly the difference in fluxes or some other information supporting the presented claim could be provided. Also, the large error bars make it hard to really make such a distinguished claim.
8. L. 271: The authors only mention a "set threshold". Could this be specified? This would then also make the next sentence, specifying three different thresholds, which were all combined in the end no longer necessary. Maybe just specify the "weakest" threshold.
9. L. 280: The authors claim that it is "evident that [the] higher the $Z_{LLJ}$, [...][the] higher the hub-height wind speed". However, Fig. 4c shows that a maximum LLJ height is observed for hub height wind speeds of 13 m/s with a slight drop-off thereafter.

_Technical Corrections:_

10. L. 140: Here, the unit GWh is written out, which is not necessary and is also not done for other units, e.g., meters or Megawatts.
11. L. 142: Here, it sounds like with "the millions of U.S. homes" all homes in the entire state/country are meant. As this is not the case, I would leave out the "the".
12. L. 156: It should be "correct" instead of "correcting"
13. L. 503: I think here it should be "impact on wake recovery", not "of"
14. L. 511: "Gravity waves enhance" instead of "enhances"

_Literature_

Peng, S., Yang, Q., Shupe, M. D., Xi, X., Han, B., Chen, D., Dahlke, S., and Liu, C.: The characteristics of atmospheric boundary layer height over the Arctic Ocean during MOSAiC, Atmos. Chem. Phys., 23, 8683–8703, https://doi.org/10.5194/acp-23-8683-2023, 2023.

---

## Author Response (AR2)

Review of *"Observations of wind farm wake recovery at an operating wind farm"* by Krishnamurthy, R., Newsom, R., Kaul, C., Letizia, S., Pekour, M., Hamilton, N., Chand, D., Flynn, D. M., Bodini, N., and Moriarty, P.

The revised manuscript analyses the vertical profiles of the vertical momentum flux and vertical wind speed within a wake induced by a large wind farm in the US Great Plains. In their paper, the authors distinguish between several meteorological parameters, including atmospheric stability, boundary layer height, presence of LLJ events and extreme veer and shear occurrences. Further, the authors provide an exemplary extreme case with a very high downward flux in the wake, possibly induced by the presence of a gravity wave. The results show a clear dependence of vertical momentum flux and wind speed deficit on the prevailing atmospheric stability regime, as well as on the presence of extreme events, such as LLJs and in one particular case a gravity wave. Further, observations suggest, that the wind farm's effects are present throughout the entire atmospheric boundary layer, even far above the rotor plane. Thus, the manuscript addresses internationally relevant questions of importance for the scientific community within the scope of the journal.

From my point of view, the language used in the presented manuscript is very nice and the writing style is easy to follow. The chosen title is concise and represents the content of the paper quite well. The authors provide a very thorough and informative literature overview and separate their work from previous research.

Within the introduction of the paper, the objective statement is formulated very vague. Instead, I would suggest that the analysis of the wake properties is directly included (cf. comment #7).

The structure of the revised manuscript is now easier to grasp and follows a clear storyline. Sections that do not directly add value to the main objective of the paper are now moved to the appendix and provide valuable additional information on the measurements and post-processing.

Considering this and the comments presented in the following, I would recommend the manuscript for a minor revision.

We thank the reviewer for their valuable comments and suggestions. We have addressed all the comments in the updated manuscript. Our responses to the reviewer comments in black are provided in blue below.

*General comments:*
1. The authors jump between abbreviations and the written version of LLJ and low-level jet. (e.g. L. 10, 14, 16, 24, 278, etc.)

These have been made consistent in the updated manuscript.

2. Sometimes, adding a "the" would lead to increased readability, e.g. L.236: "[The] larger the vertical momentum flux, [the] faster the wake […]" or L. 241-242: "the impact of conventional updrafts or downdrafts on [the] propagation of wakes"

Thanks for the comment. We fixed the two instances mentioned by the reviewer but have also identified others in the manuscript. Please see the revised manuscript.

3. In Fig. 3 error bars are given for the median profiles, while they are missing in Fig. 5 and following. Is there any specific reasoning behind this?

Yes, we felt that by adding the error bars, although they provide valuable information, the median profiles without the error bars are cleaner to understand the trends shown in the paper. Especially when in Figure 5 and following, we have 4 vertical profiles shown in one panel. The error bars would significantly clutter the figure and make them less readable.

*Specific comments:*
4. L. 28: You only mention mesoscale simulations here, but e.g. in Schneemann et al. 2020 the authors observed them with scanning offshore lidars.

We agree, the authors have included "…and offshore observations…" in the updated manuscript.

5. L. 53: The formulation "ABL is lower than 300m" suggests that this is always the case. However, as per my knowledge, even in stable conditions boundary layer height can exceed 300m (e.g. Peng et al. 2023).

Thanks for this comment, we agree, and have made the below change to the text.

"…and in offshore or stable atmospheric conditions the ABL can be lower than 300 m."

6. L. 209: Here, the authors claim that "sufficient" data is available. It would be helpful to know, how much (e.g. in hours or No. of measurements) that is.

Thanks for pointing this out, we have mentioned the number of measurements available from the southerly wind direction chosen in this manuscript. The revised sentence is provided below.

"Since the wind directions are predominantly southerly, sufficient data is available (1490 10-min samples) to accurately estimate the mean trends of momentum flux during specific atmospheric conditions."

7. L. 229: The authors claim, that larger momentum flux deficits for near surface areas are observed for unstable and neutral conditions. However, I would argue that during stable conditions, based on the provided figures, the momentum flux deficit is larger than for neutral conditions and also compared to unstable conditions. At larger heights it then seems as if momentum flux deficits are larger during unstable conditions. Maybe a clearer picture containing directly the difference in fluxes or some other information supporting the presented claim could be provided. Also, the large error bars make it hard to really make such a distinguished claim.

In this sentence we are only discussing about the large vertical momentum fluxes observed in neutral and unstable conditions compared to stable conditions, and not the momentum flux deficits. But to the reviewers point, the momentum flux deficits are larger during unstable conditions at higher altitudes but larger during stable conditions at lower heights (see Figure below just for the reviewer, and we feel is evident from the existing figures in the paper). We have added the below sentence to the manuscript at Line 240: "Overall, momentum flux deficits are greater during unstable conditions at higher altitudes, while they are larger during stable conditions at lower heights."

[Figure]

Figure: Median momentum flux difference between the upwind and downwind during unstable, stable, and neutral conditions. Not shown in the updated manuscript.

8. L. 271: The authors only mention a "set threshold". Could this be specified? This would then also make the next sentence, specifying three different thresholds, which were all combined in the end no longer necessary. Maybe just specify the "weakest" threshold.

The weakest drop off threshold of 5 ms$^{-1}$ is mentioned in the updated manuscript. The updated sentence is provided below:

"The definition is based off two criteria, 1) wind speed maximum (*i.e.,* LLJ nose) is observed within the lowest 2-km and is greater than at least >10 ms$^{-1}$, and 2) wind speed drop-off above the jet-nose is observed and above a set threshold (at a minimum > 5 ms$^{-1}$)."

9. L. 280: The authors claim that it is "evident that [the] higher the $Z_{LLJ}$, [...][the] higher the hub height wind speed". However, Fig. 4c shows that a maximum LLJ height is observed for hub height wind speeds of 13 m/s with a slight drop-off thereafter.

The sentence has been rephrased to: "It is evident that, up to near rated wind speed (approximately 13 ms$^{-1}$), a higher $Z_{LLJ}$ results in a higher jet nose wind speed and a higher hub-height wind speed."

*Technical Corrections:*
10. L. 140: Here, the unit GWh is written out, which is not necessary and is also not done for other units, e.g., meters or Megawatts.

We agree and the acronym is not spelled out in the updated manuscript.

11. L. 142: Here, it sounds like with "the millions of U.S. homes" all homes in the entire state/country are meant. As this is not the case, I would leave out the "the".

The word "the" is removed in the updated manuscript.

12. L. 156: It should be "correct" instead of "correcting"

Thank you and the typo is now corrected in the updated manuscript.

13. L. 503: I think here it should be "impact on wake recovery", not "of"

The typo is corrected in the updated manuscript.

14. L. 511: "Gravity waves enhance" instead of "enhances"

The typo is corrected in the updated manuscript.

*Literature*
Peng, S., Yang, Q., Shupe, M. D., Xi, X., Han, B., Chen, D., Dahlke, S., and Liu, C.: The characteristics of atmospheric boundary layer height over the Arctic Ocean during MOSAiC, Atmos. Chem. Phys., 23, 8683–8703, https://doi.org/10.5194/acp-23-8683-2023, 2023.